# Research on Design, Calibration and Real-Time Image Expansion Technology of Unmanned System Variable-Scale Panoramic Vision System

**DOI:** 10.3390/s21144708

**Published:** 2021-07-09

**Authors:** Xiaodong Guo, Zhoubo Wang, Wei Zhou, Zhenhai Zhang

**Affiliations:** School of Mechatronical Engineering, Beijing Institute of Technology, Beijing 100081, China; guoxiaodong0210@163.com (X.G.); wangzhoubo283@163.com (Z.W.); hbjky_zzr@163.com (W.Z.)

**Keywords:** single-view, catadioptric reflection, unmanned systems, panoramic calibration, panoramic deployment

## Abstract

This paper summarized the research status, imaging model, systems calibration, distortion correction, and panoramic expansion of panoramic vision systems, pointed out the existing problems and put forward the prospect of future research. According to the research status of panoramic vision systems, a panoramic vision system with single viewpoint of refraction and reflection is designed. The systems had the characteristics of fast acquisition, low manufacturing cost, fixed single-view imaging, integrated imaging, and automatic switching depth of field. Based on these systems, an improved nonlinear optimization polynomial fitting method is proposed to calibrate the monocular HOVS, and the binocular HOVS is calibrated with the Aruco label. This method not only improves the robustness of the calibration results, but also simplifies the calibration process. Finally, a real-time method of panoramic map of multi-function vehicle based on vcam is proposed.

## 1. Introduction

The key technologies of unmanned systems can be divided into four parts [1]: environment perception, precise positioning technology, decision making and planning, and control and execution technology. Among them, environment perception is the premise and the most important technology to solve other key technologies. In the environmental perception technology, the common vision systems are widely used as an indispensable passive sensor system with low financial cost. However, restricted by the optical principle of perspective lens, it can only observe the local area of the environment, so that the amount of environmental information obtained by vehicles is very limited, Moreover, the depth information of the environment cannot be obtained only by the single common vehicle vision systems. In order to meet the needs of large scale, large field of view, and integrated imaging of vehicle vision systems, panoramic systems came into being.

The concept of “panorama” was first put forward in the field of European art, and then engineers gradually explored the advantages of this concept. So far, the application of panorama has changed from art to engineering, and has gradually been sought after by all walks of life. Donald W. Rees, an American scholar, proposed omni directional vision sensor (ODVS) [2] in 1970 and applied for a patent. Since then, the research on panoramic vision systems has been further expanded by Yagi, Hong and Yamazawa in 1991, 1994, and 1995, respectively [3,4,5]. Panoramic vision systems have the advantages of large field of view, integrated imaging, imaging symmetry, rotation invariance [6], especially in the fields of visual navigation, panoramic vision slam [7], visual odometer, active vision, unmanned systems, space field of taking high-definition panoramic images on the moon, panoramic vision systems monitoring, and underwater detection [8]. According to their different components, panoramic vision systems can be divided into pan-tilt rotating panoramic vision systems, fisheye lens panoramic vision systems, multi-camera splicing panoramic vision systems [9], catadioptric panoramic vision systems, and panoramic annular optical vision systems [10]. Compared with other conventional methods such as large field of view imaging, catadioptric panoramic imaging systems has great advantages in miniaturization, structural flexibility, low cost, and real-time acquisition.

Since the year 2000, IEEE-ICOIM has held seminars on panoramic imaging for many years that mainly focus on articles on catadioptric panoramic imaging [11]. As a cross-discipline of computer vision and optics, catadioptric panoramic imaging still has many theoretical and technical problems to be solved urgently. In particular, the imaging properties, calibration methods, distortion correction of panoramic images, panoramic image expansion, stereo matching of panoramic images, and theories and methods of stereo reconstruction of single-viewpoint catadioptric panoramic vision systems all need further research. Among them, the single-view hyperboloid catadioptric panoramic vision systems have the advantages of good systems design flexibility, good integrated imaging effect, large field of view, and high performance in real-time imaging, and has been gradually applied in the field of unmanned systems technology.

## 2. Background and Related Work

The author have summarized the previous research work from four aspects: panoramic imaging model, panoramic systems calibration, distortion correction of panoramic image, and panoramic image expansion.

### 2.1. Panoramic Imaging Model

The standard vision systems are described by the pinhole imaging model [12], and modeled by perspective projection. However, there are some image systems that suffer from high-distortion artifacts, such as panoramic vision systems, which cannot be described by the pinhole imaging model. Due to the fact that the panoramic vision systems need to consider the reflection of the fisheye camera lens, when modeled, these systems cannot be described by the traditional pinhole camera model. For the calibration of the panoramic vision systems, many different imaging models have emerged, such as: pinhole imaging model, central refraction spherical unified model, pixel ray model, radial distortion model, fisheye camera model, Taylor series model, and MEI camera model, and there are many different calibration methods for the same imaging model.

#### 2.1.1. Pinhole Imaging Model

The standard camera, as an image sensor, is an important part of vision measurement, and can perform the mapping of two- and three-dimensional environmental points. The imaging model of ordinary cameras is often nonlinear, but because the field of view angle of ordinary cameras is very small, it can be simplified as an ideal model (linear model) for pinhole imaging, which is equivalent to the object being shot onto the CCD or CMOS sensor through the pinhole. The pinhole imaging model is a kind of mapping that projects all the environmental points onto the image plane through the center point of the camera’s optical axis, which can be described as a set of virtual sensor elements [13]. Each virtual sensor element contains its inherent optical and geometric properties, so a certain mapping relationship is established between environmental points and image pixels.

There are three main methods to perform the calibration of the pinhole imaging model: The first one is based on a two-dimensional grid. The most representative ones are Tsai [14] two-step method proposed in 1987 and Zhang’s [15] calibration method in 2000. They use a calibration board composed of two-dimensional squares for the calibration method. They collect different pose images of the calibration board, extract the pixel coordinates of corners in the image, calculate the initial values of internal and external parameters of the camera through homograph matrix, and estimate the distortion coefficient by the nonlinear least squares method. Finally, the maximum likelihood estimation method is used to optimize the parameters. This method is easy to operate and offers high enough precision for most cases such as in the case of only two-dimensional calibration board, the systems to be calibrated are not easy to move. The second method is to control the camera to perform directional motion through the active systems [16].

Based on the previously mentioned systems, and using the control platform to make the camera perform specific movements and take multiple groups of images, one can combine the image information and the known displacement changes in order to solve the internal and external parameters of the camera. The third method is camera self-calibration. The camera self-calibration method presents some advantages such as strong flexibility and online calibration. The self-calibration method based on Kruppa’s [17] proposed method consists in establishing the constraint equation about the camera’s internal parameter matrix through the quadratic curve, and use at least three pairs of images to calibrate the camera. The length of the image sequence will affect the stability of the calibration algorithm, which cannot guarantee the infinite plane in the projective space.

#### 2.1.2. Unified Imaging Model of Homograph Matrix Central Catadioptric Sphere

In 2000, Christopher Geyer and Kostas Daniilidis et al. [18] proposed the unified projection model for central catadioptric cameras, which can directly and naturally observe the invariance of these projections, it is pointed out that any catadioptric projection and standard perspective projection are equivalent to a projection map. The catadioptric camera model includes hyperboloid mirror camera model, parabolic mirror camera model, and ellipsoid mirror camera model. Kang, S. B et al. [19] proposed a self-calibration method for the unified imaging model of central catadioptric camera. Because it does not need any special calibration mode, nor does it need any camera motion or scene geometry knowledge, it is very a convenient method to perform the calibration. In 2002, the Svoboda and Pajdla et al. [20] used the spherical uniform model of the central catadioptric to derive the constraints of the corresponding image points, and then extended the classic Epipolar geometry of the perspective camera to all central catadioptric cameras. Finally, the geometric constraints of the outer pole of all types of centers catadioptric cameras were derived, which provides a new idea for the calibration of panoramic cameras in the future. In 2004, Micušık, B et al. [21] extended the spherical unified model of central refraction, and extended the Fitzgibbon method for estimating the radial distortion. It not only obtained the initial estimation of camera model and motion state, but also obtained more accurate 3D reconstructed images. In 2005, João P. Barreto et al. [22] optimized the spherical uniform model of the central catadioptric reflection. The optimized projection model only needs three-line graphs to calibrate any central catadioptric systems. In 2011, Luis Puig and Yalin Bastanlar [23] proposed a DLT (direct linear transformation) calibration technique based on the spherical uniform model of the central refraction. In this model, each central catadioptric system is divided into two projections, one is the perspective projection from a 3D point to a single sphere, and the other is the perspective projection from the sphere to the image plane. The internal and external parameters of the systems are estimated by solving a system of linear equations. The parameter estimation is taken as the initial value of the systems parameters, and the parameters are optimized by minimizing the reprojection error.

#### 2.1.3. Pixel Ray Model

In 2005, Sturm et al. [24] proposed a novel pixel ray model, which is composed of the coordinates of the projection ray, the ray and the mapping between pixels. The model is suitable for any type of camera. In 2007, Olivier morel and David Fofi et al. [25] proposed a polarization image calibration method based on this model. As it only needs optical equipment, and does not require image processing and calibration mode, non-professionals can also benefit and perform a “one click” calibration, and the model is suitable for any shape of mirror.

#### 2.1.4. Radial Distortion Model

In 2006, Kannala, J and Brandt, S.S et al. [26] proposed a general radial distortion model, which is suitable for the calibration of fisheye lens, traditional lens, and wide-angle lens, and proposed a calibration method based on the observation plane calibration pattern to estimate the model parameters, and its calibration accuracy can reach 5 pixels.

#### 2.1.5. Fisheye Camera Model

The application of fisheye lenses in wide viewing angles is relatively convenient, but due to the lack of accurate, universal and easy-to-use calibration procedures, their targets and purposes for measurement are greatly limited [27]. The projection model of fisheye lens is divided into four types: equidistant projection, equal solid angle projection, stereo projection and orthogonal projection [28].

In 2007, Jonathan Courbon et al. [29] proposed a general model of fisheye lens, which is composed of perspective projection onto a virtual single sphere and image plane, and has been widely used in practical applications.

#### 2.1.6. Taylor Series Model

In 2006, Scaramuzza, D. et al. [30,31,32] proposed a model that uses Taylor polynomials to represent the unified projection model of the central catadioptric camera and the fisheye camera model. In order to overcome the lack of knowledge for fisheye camera parameter models, a Taylor polynomial model is used to analyze catadioptric and dioptric cameras. The panoramic vision systems based on Taylor polynomial model uses a checkerboard as a calibration object and employs the toolbox of Scaramuzza [33] to calibrate. In 2009, D. Schneider et al. [34] proposed a basic geometric model extended by the standard polynomial correction method to simulate the radial symmetric lens distortion to achieve sub-pixel accuracy, which has achieved good results in most fisheye lens applications. In 2015, Seffen urban and Jens Leitloff et al. [35] proposed a generalized camera model based on Scaramuzza et al., which improved the method of calibrating wide-angle, fisheye lens and panoramic imaging systems. The improved calibration program has better robustness, its accuracy is seven times of the original, and the necessary calibration steps are reduced from five to three.

#### 2.1.7. Mei Camera Model

In 2007, Christopher Mei and Patrick rives et al. [36] proposed a calibration method for single-view panoramic vision systems. This model is based on an accurate theoretical projection function, by adding parameters to the function to simulate the real-world error, and estimate the parameters of catadioptric panoramic camera and fisheye camera.

#### 2.1.8. Other Camera Models

In 2007 Frank, O and Katz, R et al. [37] proposed a new camera model composed of distortion center, lens distortion function and sensor relative to the internal position of the lens, and extended the Scaramuzza algorithm. Compared with the Scaramuzza algorithm, the model improved the calibration accuracy from 5 to 3 pixels. In terms of the complexity and stability of the calibration process, this model is presenting a good compromise to the general traditional camera model.

### 2.2. Panoramic Systems Calibration

Due to the complexity of the camera projection model and the difficulty of assembling high-quality sensors, the accurate calibration of panoramic vision systems is more challenging than in ordinary vision systems [38]. Scale information is extracted from the environment, and panoramic systems calibration is the first necessary step to determine the mapping relationship between the environment point coordinates and image pixel coordinates, and to correct these errors in the application of panoramic vision systems. The internal and external parameters of the systems can be obtained by calibration. The internal parameters describe the projection relationship of the object point from the camera coordinate systems to the pixel coordinate systems, mainly including focal length, principal point, pixel aspect ratio and skew. The external parameter describes the rotation and translation of the environment point from the world coordinate systems to the camera coordinate systems.

The calibration method based on the calibration object consists of using an object with known information on its shape and size, detecting some image features of the calibration object (within the calibration picture), and afterwards establishing a connection between the real information of the object and the information extracted from the picture. However, the general image features, such as SIFT feature points [39], Harris corner points, Camry edge, and so on, are usually local and cannot use the global information of the image, resulting in their inaccuracy and robustness. Moreover, the process of extracting such local features is usually very cumbersome, and the accuracy of extraction cannot be guaranteed. Therefore, extracting local features of such images is still one of the bottlenecks of camera calibration methods. The calibration methods of panoramic vision systems have been extensively studied in recent decades. These calibration methods can be roughly divided into six categories according to the types of calibration objects.

#### 2.2.1. Calibration Based on Point Projection

The method based on point projection uses a calibration mode with control points whose 3D world coordinates are known. These control points can be corners, points, or any feature that can be easily extracted from the image. The internal and external parameters of the panoramic vision systems are estimated through the position of the control point and the projection position on the image. In 1971, Y.I. Abdel Aziz et al. [40] established the relationship between the environmental points and the corresponding points in the captured image, and used DLT transform to describe the imaging process of the camera linearly, but it did not take into account the nonlinear distortion of the camera.

In 2000, S. B. Kang et al. [19] developed a reliable calibration method for catadioptric panoramic systems without using any special calibration mode, camera motion knowledge, or scene geometry knowledge by using the feature consistency of paired tracking points in image sequence. In 2001, D.G. Aliaga et al. [41] used the iterative method based on 3D control point coordinates to restore the internal and external parameters (position and direction) of the non-central catadioptric panoramic systems. In 2005, Wu Y. et al. [42] proposed the invariant characteristics of projection geometry between environmental points and image points for catadioptric panoramic systems without eliminating distortion. The main advantage of using these invariant characteristics for planar reconstruction is that apart from the principal point, neither camera movement nor inherent parameters are required. In 2011, Puig et al. [43] used the projection matrix to linearly represent the projection of 3D points on the image, so as to establish the corresponding relationship between 3D environment points and 2D image points for calibration. The nonlinear optimization method is used to solve the matrix, and the internal and external parameters of the systems can be obtained from the decomposition results. In 2012, Thirthala et al. [44] proposed a multifocal tensor generated by a one-dimensional radial camera to calibrate the refraction and reflection panoramic systems. When a fourth-order tensor is used, the scene can be metrically reconstructed and the image distortion removed.

#### 2.2.2. Calibration Based on Single Image Line Projection

Many calibration methods are based on the projection of lines in panoramic images [45]. Lines are used mainly because they exist in many environments and do not require special patterns or any measurement information. These methods calculate the absolute conic Image, from which the inherent parameters of the panoramic systems are calculated.

In 1999, C. Geyer et al. [45] used images of two sets of parallel lines to find the inherent parameters of catadioptric panoramic systems and the direction of the plane containing two sets of parallel lines. In 2000, Rahul Swaminathan et al. [46] proposed that the lens distortion parameters can be obtained directly by using the image line features, without establishing the relationship between the spatial coordinate systems. In 2002, C. Geyer and K. Daniilidis [47] proposed a method for calibrating a catadioptric panoramic systems. The calibration parameters of the systems can be obtained only by the projection of three non-metric information lines in the space image. In 2003, J.P. Barreto et al. [48] used line images to calibrate the panoramic vision systems, but at least three-line images were needed to calibrate the vision systems with good accuracy. In 2004, X. Ying et al. [49] proposed a new calibration method based on geometric invariants, which provides a unified framework for the calibration of line image or spherical image. In 2004 Vasseur, P. et al. [50] used the projection of spatial straight lines to estimate the internal parameters of any single point of view refraction systems. In 2006, Vandeportaele, B. et al. [51] optimized the C. Geyer and K. Daniilidis method, and proposed a new method to calibrate the diopter camera by using one image of at least three observation lines. The method uses geometric distance instead of algebraic distance. They allow the line projecting to a straight line or arc in a unified way. In 2007, V. Caglioti et al. [52] proposed an off-axis catadioptric systems calibration method, which is different from the standard catadioptric panoramic systems. In the off-axis systems, the perspective camera can be placed at almost any position and only uses the mirror surface profile and at least one frame. The image of the ordinary space line can be used to calibrate the internal and external parameters of the systems. In 2008, F. Wu et al. [53] proposed the relationship between the projection of the space point on the observation sphere and the refraction image. According to this relationship, they estimate the parameters used to calibrate any central refraction panoramic systems.

#### 2.2.3. Object Calibration Based on 2D Calibration

Two-dimensional based calibration relies on control points within an indicator pattern. These control points can be corner points (or any other specific points), or any characteristic features that can be extracted and quantified from the image. By using an iterative method, the internal and external parameters can be inferred. Because 2D images can easily cover the whole refraction image, their methods can be calibrated accurately and their algorithms are also complex. The most representative one is Tsai’s two-step [14] and Zhang’s calibration methods using a two-dimensional square calibration board. It consists in collecting pictures of the calibration board in different positions, extract the pixel coordinates of the corner points from the picture, calculate the initial value of the camera’s internal and external parameters by using the single matrix, estimate the distortion coefficient by the nonlinear least squares method, and finally optimize the parameters by using the maximum likelihood estimation method. The method is operationally simple and offers good precision in most applications, and has gradually developed into a popular calibration method in computer vision. In 2002 H. Backstein et al. [54] used two-dimensional rectangular checkerboards to calibrate the fisheye camera to achieve seamless image mosaic 360° × 360°. In 2006, Luo C et al. [55] calibrated pal optical systems with 2D checkerboard mark, and the accuracy of calibration results reached the practical application level. In 2006, Kannala J. et al. [26] proposed a 2D calibration method for fisheye camera, which is used for 3D image reconstruction. In 2006, Scaramuzza D [30] and Steffen Urban [35] in 2015 proposed a method of calibrating the panoramic systems of refraction based on 2D markers. They assumed that the projection function of the image can be modeled by a Taylor series expansion, and its coefficient is estimated by solving the two-step least squares linear minimization problem. The latter is the same as the former, except that the cost function of the former is modified to obtain more accurate and stable calibration results. In 2007, Deng Xiao-Ming [56] proposed a simple 2D calibration model for the central refraction panoramic systems calibration, as it is described in the following.

Firstly, the boundary ellipse and field of view (FOV) of the catadioptric image are used to obtain the initial estimation of the eigen parameters. Then, the external parameters are initialized by using the explicit relationship between the central refractivity and the pinhole model. Finally, the internal and external parameters are refined by nonlinear optimization. This method does not require any fitting of the partially visible conic, and the projected image of the 2D calibration pattern can easily cover the entire image, so the method is simple and robust. In 2007, C. Mei et al. [36] proposed a single-view catadioptric panoramic vision systems calibration method based on an accurate theoretical projection function sphere model, which added some additional radial and tangential distortion parameters to consider the real-world error of the plane mesh. In 2009, S. Gasparini et al. [57] proposed a new calibration technique of catadioptric panoramic camera based on a planar grid image. This method is calibrated by more than three plane grid images, and finally the central refractive panoramic systems parameters are recovered from the intermediate focal speckle. In 2011, Z. Zhang [58] and 2012, Zhang Dong Zhang et al. [59] proposed a panoramic vision systems calibration method based on a new 2D image feature—TILT (transform invariant low rank texture). Compared with the traditional calibration method, this method has the advantages of simple operation and wider application range. In 2016, Zhang Zhu et al. [60] proposed a hyperboloid catadioptric panoramic camera calibration method based on Halcon. During the calibration process, only the relevant parameters of the calibration plate need to be known, and there is no need to have a prior knowledge of the scene, nor special devices and equipment. This calibration method has high accuracy and high speed. In 2017, Hu Shi hui et al. [61] calibrated the vehicle vision systems by extracting the features of multiple symmetrical 2D checkerboard markers (TILT) at the same time, and the symmetry of 2D markers improves the calibration accuracy to a certain extent.

#### 2.2.4. Object Calibration Based on 3D Calibration

The calibration method based on 3D calibration object has certain requirements for the calibration object, such as the geometry and three-3D position of the 3D calibration object are known, and it has some features that can be accurately detected (for example, the grid with black-and-white Square). In 2004, Vasseur P. et al. [50] proposed a method to obtain the equivalent sphere model and camera intrinsic parameters of panoramic systems by detecting 3D lines in the image. In 2005, R. Hartley et al. [62] proposed a method based on 3D checkerboard markers to simultaneously calibrate the radial distortion function and other internal calibration parameters of the camera. In 2007, the first mock exam of the center catadioptric panorama systems were put forward by C. Toepfer et al. [63] in 2007, which could select multi-point, 2D calibration material or calibration material. In 2011, L. Puig et al. [23] proposed a calibration technique suitable for all single-view catadioptric panoramic vision systems, which obtains the general matrix of the systems through 3D markers, and then completes the parameters of the panoramic vision systems.

#### 2.2.5. Self-Calibration

Compared to the previous method, the self-calibration technique does not need any calibration object. It only uses the geometric or motion information of some specific objects from the picture in order to perform the camera calibrate. The self-calibration method has the following advantages: 1. Calibration can be carried out anywhere. 2. Is less dependent on specific scene structure, and there is no need to set a calibration mode in the 3D world. However, the self-calibration method has not received too many academic attentions due to the low accuracy of the calibration results. Self-calibration methods mainly include techniques based on orthogonal directions, usage of pedestrian’s information within a video, and on the analysis of three-dimensional object structures, such as buildings, within picture. In 1992, O.D. Faugeras et al. [64] proposed a complete method of camera self-calibration. Compared with the existing methods, it does not need calibration objects with known 3D shapes, but only points matching in image sequences. In 1994, Hartley R.I [65], 1995, G. P. Stein [66], 2010, S. Ramalingam [67], and 2011, F. Espuny [68] used camera rotation and translation calibration methods to perform the self-calibration of panoramic vision systems. In 1997, Trigg’s b et al. [69] is the first time to used quadric surface method for camera self-calibration. The Trigg’s method is more accurate than that based on Cruppa’s equation. In 1998, D. Liebowitz et al. [70] and in 1990, Caprile, B. et al. [71] proposed the vanishing point method based on orthogonal directions to self-calibrate the panoramic vision systems. In 2000, S.B. Kang et al. [19] used the consistency of paired tracking point characteristics to calibrate the systems. In 2001, Viola and Paul et al. [72] calibrated the panoramic vision systems by constraining the corresponding points in multiple images. In 2003, B. Micušık [73] used the corresponding points of different viewing angles and the least squares method based on Epipolar constraints to perform systems self-calibration. Maybank, S.J et al. [74] proposed a method of self-calibration of the panoramic vision systems based on pedestrian in video in 2006. Strecha, C. et al. [75] in 2008 and Yasutaka Furukawa [76] in 2009 used the 3D buildings in the picture to perform the self-calibration of the panoramic vision systems. In 2012, Dazhou Wei [77] used the Homograph to calibrate the cataracted panoramic camera, which is more robust than the traditional self-calibration method. In 2013, Rahul Swaminathan and sheer K. Naya [78] proposed a method to calibrate panoramic vision systems by finding distortion parameters that map image curves to straight lines without using any calibration object. This method has been widely used in wide-angle camera clusters. In 2019, Choi, K.H. et al. [79] used V-type, an A-type, a plane-type, and room-type test object to conduct the self-calibration test of the panoramic vision systems. Finally, a self-calibration method for the precise panoramic vision systems applied to V-type test piece was proposed. The root mean square residual of this method is less than 1 pixel.

#### 2.2.6. Other

In 2006, Morel, O. et al. [80] proposed a three-dimensional automatic detection system based on polarization analysis, this method can solve the problem of three-dimensional detection of highly reflective metal objects. In 2007, D. Fofi et al. [25] proposed a new and effective calibration method of catadioptric panoramic vision systems based on polarization imaging. In 2013, Ainouz, et al. [81] proposed a non-parametric definition method in pixel neighborhood of the refractor image, this method is nonparametric and enables to respect the catadioptric image’s anamorphosis. In 2017, Yujie Luo [82] proposed a compact polarization-based dual-view panoramic lens to solve the two major limitations of the conventional PAL structure: Large front annular lens and lack of forward view.

### 2.3. Distortion Correction of Panoramic Image

The panoramic image exhibits serious nonlinear distortion due to the involvement of a quadratic reflection mirror. The intuitive feeling is that the straight line in the image will become a curve, and the resolution of the edge of the image will be less than the resolution of the image center. If this type of image is directly used for object detection, image recognition and other aspects will cause huge errors, so it is necessary to correct the distortions in the panoramic image.

The image produced by panoramic systems contains a lot of radial distortion, tangential distortion, internal reference error, image plane distortion and inherent proportion change. Although the complex distortion model can improve the calibration accuracy under certain conditions, because it is related to the optimization algorithm of nonlinear equation, the introduction of too many variable parameters will increase the correlation of the equation, leading to the instability of the settlement process. The general principle is to simplify the distortion model as much as possible within the accuracy range. The reason of panoramic image distortion is that the reflector is a curved surface, so as long as the reflection law of the curved surface is known, the panoramic image can be corrected. In recent years, there are endless papers on the distortion correction of panoramic lenses. According to the correction methods, they are mainly divided into two categories: polynomial-based distortion correction and non-polynomial-based distortion correction [83].

#### 2.3.1. Image Distortion Correction Based on Polynomial Functions

From the perspective of an embedded application, polynomial method is easy to mention. Because polynomial is simpler than trigonometric function and logarithm operation, it can reduce the amount of operation. In 1971, D. C. Brown et al. [84] proposed to use Gaussian distortion function to represent radial distortion and eccentric distortion in order to obtain the highest accuracy. Experiments confirmed the correctness of the theoretical research. The experimental results show that the theoretical research is correct. In 1980, Slama [85] proposed that the singular polynomial was used to represent radial distortion, and then it was widely used in distortion correction. In 1994, G. Wei [86] and J. Heinkel [87] in 1997, implicit rational polynomials were used to approximate inverse functions. In 1994, Shishir Shah et al. [88] obtained the polynomial relation between the global coordinate point and the corresponding image surface by Lagrange estimation method, and verified the rate of distortion correction. In 1995, Frédéric Devernay [89] used polynomial to represent tangential distortion, which has higher accuracy than traditional distortion correction methods. In 2001, Fitzgibbon et al. [90] introduced the “division model”, which was usually used with circle fitting to correct distortion. The method is more accurate in describing small distortion, but it is not very good for the lens with large field angle and fisheye lens. In 2004, J. Mallon [91] deduced the inverse function of radial distortion by first order Taylor expansion, and then reconstructed the radial distortion parameters. The accuracy of the function can reach sub-pixel precision compared with the same level of accuracy. In 2005, M. Ahmed et al. [92] proposed a method to improve the accuracy of distortion correction by fixing the distortion center in the appropriate position (for example, image center), and then using two eccentric distortion coefficients to compensate for the reasonable deviation between the center and the real position. In 2006, Kannala et al. [26] proposed a universal odd-degree polynomial to represent the distortion model of ordinary lenses based on the radial distortion symmetry characteristics of the panoramic vision systems, and gave a complete calibration method. In 2010, Hughes, C. [93] used a method called polynomial fisheye transform (PFET) to describe the distortion of the fisheye lens. In 2017, Shihui Hu et al. [61] used the seventh-order odd order polynomial to fit the nonlinear radial distortion in panoramic vision systems. This polynomial can improve the calibration accuracy of panoramic image.

#### 2.3.2. Image Distortion Correction Based on Non-Polynomial Functions

Most of the non-polynomial distortion correction methods are based on the analysis method. The idea is to establish the stable relationship between the image to be corrected and the corrected image, which is very important for the image correction of panoramic vision systems. In 1997, B. Prescott et al. [94] proposed an on-line detection method to determine the distortion parameters of linear radial lens. This method does not need to determine the corresponding relationship between environmental points and image points. In 1999, Gaspar J. et al. [95] used the distance mapping table between the ground plane and the image plane to correct the image distortion. In 2001, Zhang Zheng you [15] used the camera to photograph the chessboard plane in different directions to achieve the purpose of calibration, which had a good effect. “Zhang’s calibration method” was widely used in distortion correction. In 2001, M. T. Ahmed et al. [96] proposed a method to correct the distortion of line image by optimizing the use of nonlinear search technology to quickly find the best distortion parameters and straighten these lines. In 2002, K. Daniilidis et al. [18] transplanted the distortion correction algorithm designed by pinhole model to panoramic image. In 2003, Ishii C. et al. [97] used the “stereo model” to correct part of the panoramic image and get the straight-line projection image. At the same time, the interpolation based on image density can solve the problem that the corrected image is larger than the original image. In 2003, Zhejiang Qiu et al. [98] applied photographic invariance to correct panoramic images. The advantage of this method is that it does not depend on lens parameters and has high accuracy. In 2004, Zeng J. Y. [99] modified the pinhole imaging model and established a panoramic vision systems model with secondary radial distortion to eliminate the distortion of the systems to the horizontal scene.

In 2005, M. Ahmed et al. [92] proposed a fully automatic method based on the Least Median of Squares (LMedS) estimator for non-metric calibration of lens distortion. In 2008, Liqun Liu [100] and Weijia Feng [101] used the function method to correct the distortion parameters of the panoramic vision systems, which improved the accuracy and real-time performance compared with the previous distortion correction. In 2008, Xiao Xiao et al. [102] applied the spherical perspective projection model to the correction of panoramic image according to the characteristics of panoramic systems, which can better solve the problem of image distortion.

In 2008, Qunliu Li et al. [103] proposed a simple method to correct panoramic image distortion by using support vector machine (SVM) instead of ordinary correction model. In 2009, Carroll et al. [104] used human–computer interaction to artificially establish distorted line segments, and then used grid based nonlinear optimization technique to obtain the corrected image. This method introduces the human control of the environment, allows the change of local projection, and has a good effect. In 2011, Yu Yang et al. [105] used parameter variable Gaussian filter to correct the sawtooth and ladder distortion after panoramic expansion, and carried out real-time planar expansion of video stream from reflective panoramic camera, and achieved good results. In 2011, J. Maybank s [106] simplified the task of line detection to the task of finding isolated peaks in Sobel image according to the distortion characteristics of space line projection. Based on this task, the Fisher Rao metric was proposed to correct image distortion in parabolic mirror and telecentric lens systems. In 2012, K. Kanafani et al. [107] corrected the distortion of panoramic vision image by minimizing the characteristic value. In 2013, Ren, Xiang et al. [108] used an ADMM convex optimization method to correct image distortion based on the TILT feature. In 2014, Clapa and Jakub [28] proposed a method to correct the fisheye lens with lens equation, and improved them so that they can be applied to the field of view beyond 180°. In 2014, on the basis of Carroll’s research, Yanyan Huang et al. [109] used the corner coordinates of chessboard instead of the line feature information as the input of the correction model, and linearized and optimized the objective function after modeling. The experiments show that the method can realize the automation of distortion correction and greatly improve the efficiency of the systems. In 2014, Wu Y.H. et al. [110] established accurate geometric invariants from 2D/3D space points and their radially distorted image points, constructed a criterion function and designed a feature vector for evaluating camera lens to evaluate imaging systems, and calculated tangential distortion with infinite norm of feature vectors. In 2015, Tang [111] proposed a parameter domain mapping model based on systems prior information and catadioptric geometry to accurately express the distorted visual information. The model can directly realize adaptive distortion correction in the neighborhood of the target image according to the measured image radial distance. It is proved that the distortion neighborhood in panoramic image follows the non-linear mode. In 2015, He, Y. et al. [112] used the method of the double precision model to map the fisheye image to the spherical surface through orthogonal projection, and transformed it into horizontal longitude and vertical longitude coordinates to realize the distortion correction of panoramic image. In 2017, Shi Hui Hu et al. [113] used the TILT image correction method of sparse Bayesian learning to correct the image distortion. This method has the advantages of stronger robustness and higher correction accuracy. In 2018, Wang Zhi et al. [83] proposed two improved panoramic image correction methods according to different application environments for distortion correction of panoramic images: One is the linear projection expansion method based on the lens equation, which can correct the image of the partial viewing angle into a distortion-free linear projection image, and finally expand the method to correct the panoramic image with a viewing angle of over 180°; another method is the improved equal arc-length panoramic expansion method. The author innovatively applies the lens equation to the equal arc-length panoramic expansion method. The corrected image reflects more effective information than the traditional method, and can get good results. In 2020, X. Zhang et al. [114] calculated the image distortion by convolution neural network, and performed high-quality geometric distortion correction on the image and make it clearer. However, the convolution neural network imaging model can only be approached the ideal imaging infinitely, and the systems error is inevitable.

### 2.4. Panoramic Image Expansion

The panoramic image obtained by panoramic vision systems is a circular panoramic image, which cannot be directly used to synthesize 3D images suitable for viewing. Therefore, before stereo imaging, it is necessary to expansion the panoramic image to get a panoramic two-dimensional image that conforms to the visual habits of the human eye. The panoramic image expansion algorithm is divided into the following five categories: cylindrical expansion method, perspective expansion algorithm, concentric circle approximate expansion method, optical path tracking expansion method, and look-up table method.

#### 2.4.1. Cylinder Expansion Algorithm

The principle of cylindrical expansion image viewing is used to map the points on the panoramic image to a cylindrical surface centered on the axis of the mirror according to the optical and geometric relationship of the mirror panoramic imaging. The expansion process strictly follows the imaging model and optical principle of panoramic vision, so the expansion image not only conforms to the vision habits of the human eye, but also retains a lot of space mapping information. It is widely used in areas such as robot navigation that require highly accurate depth information extraction. The cylindrical expansion algorithm has the advantages of integrated panoramic expansion, good visualization of the expansion image, small distortion of the expansion image, suitable for single 360° panoramic image expansion, and good scene information retention. The disadvantage of the expansion diagram is that it still has a certain distortion in the longitudinal direction, the expansion algorithm is more complicated, and the systems resources are large, and it is not suitable for an implementation on embedded systems with limited resources. The cylindrical expansion algorithm needs to know the camera focal length, hyperboloid equation, and other internal parameters, as well as the relationship with the camera position and other external parameters in the process of expansion, and these parameters are not fixed in some systems; thus, in theory, the cylindrical expansion method of panoramic images is difficult to apply in real-time systems.

In 2006, Yunfeng Ling et al. [115] optimized the cylindrical expansion algorithm of panoramic image theory, and the optimized algorithm met the required one-dimensional linear transformation, one-dimensional nonlinear transformation, and two-dimensional transformation reduction algorithm based on the cylindrical expansion algorithm. In 2010, Jinguo Lin et al. [116] proposed a panoramic image center location method, which applies bilinear interpolation method to image sampling to realize the small distortion solution of hyperboloid catadioptric panoramic imaging cylindrical restoration. In 2011, Zhi Peng Chen, et al. [117] adopted an improved adaptive interpolation region interpolation method to solve the edge blur problem of traditional cylindrical expansion. In the same year, Hongjun Liu [118] and Shouzhang Xiao [119] proposed the equal arc length projection method to compress the large field of view, which solved the problem that the vertical field of view of cylindrical expansion cannot be very large, and retained part of the distortion. In 2011, Ying Wang et al. [120] proposed a multi-view and multi-form panoramic expansion algorithm based on cylindrical expansion. It includes global observation and plane expansion based on spherical coordinate system, ring image expansion based on cylindrical coordinate system and polyhedron plane expansion based on rectangular coordinate system. In 2012, Dazhou Wei et al. [77] proposed to improve the imaging quality and expansion speed of the unfolded image by using inverse perspective transformation to solve the problems of block effect and blur effect existing in the traditional cylindrical expansion algorithm. In 2013, Liangbo Ye et al. [121] achieved good results by improving the resolution of panoramic image and using forward mapping cylindrical expansion with certain interpolation algorithm. In 2018, Zhi Wang et al. [83] improved the traditional cylindrical expansion algorithm and proposed a panoramic expansion algorithm with equal arc length. The corrected image of this algorithm can provide more effective information than the traditional method and can get good expansion effect. In the same year, Chengtao Cai et al. [122] applied the panoramic cylindrical expansion algorithm to target detection of deep learning, which greatly improved the efficiency of target detection.

#### 2.4.2. Perspective Expansion Algorithms

Perspective expansion, like cylindrical expansion, is the reverse process of panoramic imaging. Using the optical and geometric principles of panoramic imaging, ray tracing is carried out, and the pixels on the panoramic image are back projected onto a specific perspective expansion plane to get the perspective expansion image. The expanded image obtained by the perspective expansion has the least distortion and is the most visually observable. However, the scope of one expansion is limited. To obtain a panoramic perspective view, multiple perspective expansions are required. At the same time, it is difficult to ensure that the scopes of the perspective expansions will not overlap, so perspective expansion is suitable for partial high-quality expansion of panoramic images. The perspective expanded image has the characteristics of small distortion and natural imaging, but its shortcomings are also obvious, that is, it is impossible to obtain panoramic information on one image. To obtain a 360° scene-expanded image, only multiple perspective operations can be performed, and the operating area cannot control the frequency accurately, so the perspective expansion is suitable for monitoring, security, and other fields that require high visual quality of the expanded image. In 2007, Yongming Feng et al. [123] used bilinear interpolation to improve the perspective expansion algorithm, and the distortion of the expansion image was greatly improved. In 2010, Lei, J et al. [124] proposed a new method to expand N-PLANE perspective panorama.

#### 2.4.3. Approximate Expansion Algorithms of Concentric Rings

The concentric circle approximate expansion algorithm divides the effective imaging area in the panoramic image into countless small circles with the center of the imaging circle as the center, and then stretches the circles one by one from the outside to the inside to have a width equal to the target expanded image small rectangular images, these rectangular images are arranged from top to bottom to form an expanded view, and the panoramic expanded view is formed by arranging rectangles with the same height as the target expanded view from the fan-shaped ring. The expansion process does not involve the imaging principle, and does not consider the parameters of the panoramic imaging mirror. The expansion operation based on the circular imaging surface of the panoramic image only carries out coordinate conversion, so it is a panoramic image expansion algorithm suitable for any mirror imaging. Although the principle of the algorithm is simple, the time consumption is short, the efficiency is high, and it is suitable for real-time imaging processing, the quality of the image obtained is far inferior to that of the optical path tracking expansion method. The concentric circle expansion algorithm ignores the non-linearity of the panoramic image information in the radial direction, so the distortion of the panoramic image is more obvious. Since the area of the ring decreases in the radial direction toward the center point, the closer of the ring area to the center point, the more obvious the distortion of the corresponding expanded image. Compared with perspective imaging, the image quality of this method is inferior to that of perspective imaging in terms of local details and shape information, especially in the maintenance of longitudinal length ratio.

In 2002, Gaspar J. et al. [125] used the concentric circle approximation expansion method to expand the panoramic image. In 2004, Wen Kai PI et al. [126] improved the concentric circle approximate expansion method with image interpolation, and the expansion accuracy was improved. In 2005, Huijie Hou et al. [127] proposed an improved concentric circle approximate expansion method, which realized the conversion from the planer cylindrical projection method (FCP) to the central projection method (CCP) used by human eyes. The panoramic image expansion effect is good, which greatly expands the application field of panoramic vision. In 2010, Ziling Ma et al. [128] combined the advantages of concentric ring approximate expansion method, optical path tracking mapping method, and look-up table expansion method according to the imaging characteristics of circular catadioptric panorama, and calculated the radial and tangential separately. The test results can meet the needs of combat robot for panoramic reconnaissance. In 2014, Xiaomi Zhu et al. [129] proposed a symmetric multiplexing panoramic image expansion method based on optical path tracking, which has the advantages of fast speed, good quality, and high efficiency compared with cylindrical expansion method and concentric circle expansion method. In 2018, Enyu Du et al. [130] made a huge breakthrough in the application of the concentric circle approximate expansion algorithm to lane line detection.

#### 2.4.4. Optical Path Tracking Coordinate Mapping Expansion Algorithms

The optical path tracking coordinate mapping method uses the principle of catadioptric panoramic imaging to perform back-projection, projecting points on the image collected by the camera to the cylindrical real space, and then tiling the cylindrical surface to obtain a rectangular panoramic image. The expansion formula and calculation difficulty are different according to the different mirror surface types, but the basic idea of panoramic image expansion is the same, and the correspondence between the pixels before and after the expansion is established based on the optical path tracking. Although the optical path tracking coordinate mapping method can obtain a high-precision, small-distorted panoramic expansion map, its algorithm is complex, computationally expensive, and has poor real-time performance. Secondly, although theoretically, continuous and distortion-free image projection transformation can be performed from the reflective surface to the cylindrical surface, because the digital image is composed of discrete and limited pixels and the pixel density of different regions is uneven before and after the transformation, the part near the center of the curved surface is transformed. The accuracy obtained afterwards decreases, and the image is blurry.

In 2002, Koyasan et al. [131] used the optical path tracking expansion method to expand the panoramic image in real-time embedded systems, which can robustly track multiple moving obstacles in complex environments. In 2007, Wei Xu et al. [132] deduced the optical path tracking expansion algorithm that expands the circular panoramic image taken by the imaging device into a cylindrical panoramic image. Aiming at the large amount of calculation and poor real-time performance in the optical path tracking expansion algorithm, he proposed that concentric circle approximate fast expansion and eight-term symmetry reuse fast expansion are two fast expansion algorithms. In 2013, Xiaoxi Zhang et al. [133] adopted the symmetric multiplexing panoramic image expansion method of optical path tracking, which solved the shortcomings of the traditional panoramic image expansion method that equality and real-time performance of the expanded image cannot be achieved, and met the real-time requirements.

#### 2.4.5. Look-Up Table Algorithm

The method of panoramic image look-up table expansion can overcome the contradiction of speed and accuracy between the optical path tracking coordinate mapping method and the concentric ring approximate expansion method. This method calculates the pixel coordinate mapping relationship between the panoramic image before and after expansion accurately in advance, and saves it in a look-up table. In this way, when the panoramic image or panoramic video is expanded in the future, as long as the corresponding pixel coordinates of each pixel in the panoramic image before expansion are found, the expansion can be carried out quickly. Because the time required for complex operations is avoided by using the look-up table expansion method for panoramic expansion, this method not only has faster expansion speed, but also has higher expansion accuracy. But because the look-up table needs a very large space for storage, in order to ensure that each frame of the image can be well expanded, it is not suitable to use the look-up table expansion method in the miniaturized and portable application systems. In 2001, Benosman et al. [134] proposed a panoramic image expansion algorithm based on look-up table method to lay the foundation for the subsequent panoramic expansion. In 2007, Zhihui Xiong et al. [135] optimized the panoramic image look-up table expansion method and proposed an eight-way symmetrical reuse strategy look-up table method, which can greatly reduce the storage space of the look-up table and increase the expansion speed of the panoramic image. In 2007, Huiying Yu et al. [136] aimed at the problems of the traditional table look-up method, such as the large storage consumption and the leakage of coordinate transformation, and optimized the coordinate transformation method in the table look-up method to meet the requirements of high speed, low storage, and real-time in PPI raster scanning display systems. In 2008, Bin Wang et al. [137] implemented catadioptric panoramic image expansion on FPGA by using the look-up table expansion method, which made the panoramic expansion speed up to 12 times higher than that based on single pixel panoramic look-up table expansion, and met the application requirements of embedded catadioptric panoramic real-time imaging systems. In 2010, Xi Chen et al. [138] combined, with the look-up table method, the panoramic image expansion algorithm based on bilinear interpolation, which improved the image expansion quality compared with the nearest neighbor sampling method. In 2015, Shitian Liang et al. [139] proposed a symmetric multiplexing inner and outer ring divide and conquer expansion algorithm after optimizing the traditional table look-up expansion method, which can improve the expansion speed of panoramic image and the quality of the expanded image. In 2018, Wei Zhu [140] adopted the block look-up table expansion method to realize the rapid expansion of catadioptric panoramic video, which greatly improved the frame rate compared with the traditional look-up table method.

Catadioptric panoramic vision systems are a new type of panoramic vision systems. Compared with the traditional multi-camera mosaic panoramic vision systems and fisheye lens panoramic vision systems, catadioptric panoramic vision systems have significant advantages in systems size, structure complexity, cost, and real-time performance. They have wide application prospects in many fields. In recent decades, catadioptric panoramic vision systems have attracted many excellent researchers to carry out fruitful research because of their important application value.

Based on the understanding of the existing research, this paper summarizes these problems and predicts the possible development direction in the future:Research on improving the resolution of catadioptric panoramic vision systems. In order to take into account the high resolution and large field of view, whether the camera can only face one side of the reflector through the off-axis of the camera, which reduces the field of view, but improves the resolution, and then cooperates with the local movement of the optical systems to achieve fast conversion between large field of view and high resolution.Research on distortion correction and real-time expansion algorithm of catadioptric panoramic vision systems. At present, the related research of single-view catadioptric panoramic vision systems mainly focuses on imaging model and systems calibration. This is only a prerequisite to solve the problem of panoramic vision. In the end, it is hoped to use the advantages of single-view catadioptric panoramic vision systems to solve typical vision problems. Thus, panoramic expansion and image distortion correction have become the main issues that need to be studied in depth in panoramic vision technology.


**Problems to be solved urgently in the catadioptric panoramic vision systems.**


Only by lengthening the baseline length can the system obtain high-precision depth information, and lengthening the baseline will inevitably increase the volume of the panoramic vision systems, which goes against the trend of systems miniaturization.The image resolution of catadioptric panoramic vision systems is unevenly distributed, the pixels near the center of the circle are more concentrated, the resolution is higher, the pixels near the edge are less, and the resolution is lower. The obstacles in the environment are in the catadioptric binocular stereo panoramic vision. In panoramic vision system, obstacles occupy different pixel values in two panoramic vision systems, which leads to stereo matching error.

According to the requirements of advanced products, this paper proposes the design index of panoramic vision systems, and uses two HOVS with the same configuration to build the vehicle binocular stereo panoramic vision perception systems of unmanned systems. The authors of this paper designed a horizontal field angle of 360°, vertical field angle of >90°, hyperboloid mirror parameters a = 48 mm, b = 44 mm, d = 80 mm, the imaging model conforms to single-viewpoint imaging, and the panorama can be expanded in real-time vehicle borne variable-scale panoramic vision system. The final design result is excellent imaging, good processability, and low price.

## 3. Architecture Design and Theoretical Analysis

### 3.1. Panoramic Systems Design

The HOVS proposed in this paper includes two single-viewpoint variable-scale hyperbolic mirror panoramic vision subsystems with the same configuration and two industrial computers, one of which is used to receive the image data collected by the panoramic vision systems in real time, and the other is used for real-time processing of image data, the data transmission between the two industrial computers is in the form of a 10 Gigabit network cable, and the two single-view variable-scale hyperboloid mirror panoramic vision systems respectively communicate with the industrial computer through a dual-channel gigabit network cable.

#### Structure Design and Depth Information Acquisition Theory of HOVS

Hyperboloid mirror module and perspective camera module

HOVS is mainly composed of two single-viewpoint variable-scale hyperbolic mirror panoramic vision subsystems with the same configuration. Each single-view variable-scale hyperboloid mirror panoramic vision subsystems are composed of a hyperboloid mirror module, a perspective camera module, a mirror mounting plate rotation module, a mirror height adjustment module, and a visual positioning module. According to the number of imaging viewpoints, reflective panoramic vision systems can be divided into single-viewpoint reflective panoramic vision systems and multi-viewpoint reflective panoramic vision systems. In order to satisfy the single-view geometric constraints of the hyperboloid mirror, the lower focus of the hyperboloid mirror should coincide with the lens of the perspective camera, so that the incident light to the upper focus of the hyperboloid mirror must be reflected to its lower focus according to the mathematical properties of the hyperboloid. Therefore, the partial position information and color information of all spatial objects on the hyperboloid mirror can be perceived in real time on the image plane, as shown in Figure 1.

Hyperboloid mirror [141] is an indispensable part of HOVS. Its vertical field of view angle and the basic size of the mirror are the design parameters that must be considered. The hyperboloid mirror is located directly above the perspective camera. According to the figure below, the upper boundary of the hyperboloid panoramic vision systems is determined by the basic parameters of the hyperboloid mirror, and the lower boundary is determined by the occlusion range of the perspective camera and the environment perception platform. For the vertical field of view *ξ,* the formula is as follows:(1)$$\left\{\begin{array}{c} \xi_{min}=arctan\left(\frac{2D}{d_{lens}}\right)\\ \xi_{max}=arctan\begin{array}{c} \left(\frac{z_{max}}{\sqrt{\left(x_{max}^{2}+y_{max}^{2}\right)}}\right) \end{array} \end{array}\right.$$
(2)$$x_{max}=\frac{d}{2},y_{max}=0,z_{max}=a,\;\eta =\frac{f_{t}-f_{f}}{f_{f}}$$
(3)$$\xi =\xi_{min}+\xi_{max}=arctan\left(\frac{2D}{d_{lens}}\right)+\;arctan(2\frac{a\sqrt{1+\frac{d^{2}}{4b^{2}}}-\sqrt{a^{2}+b^{2}}}{d})$$

From Equation (3), the vertical field angle is related to the basic parameters of the hyperboloid mirror, the lens *d_lens_* of the perspective camera, the upper focus of lens, and the distance *D* of focal plane. Under the condition of ideal single-view, *D* = 2c (where c is the focal length of hyperboloid); perspective camera lens diameter *d_lens_* = 20 mm; the opening diameter *d* of hyperboloid mirror is restricted by its processing technology and manufacturing cost. In the actual production process, as the diameter of hyperboloid mirror becomes larger, the accuracy of hyperboloid mirror becomes smaller, and the cost becomes larger. In these systems, the hyperboloid mirror surface is processed by using a composite material open mold-forming method to form an optical mirror by electroplating a reflective layer on the hyperboloid mirror surface. However, the surface area of the hyperboloid mirror will have a huge impact on the uniformity of the composite material electroplating coating, so the opening diameter of the hyperboloid mirror is determined as *d* = 80 mm, according to the actual situation of the experimental equipment. The vertical field angle diagram of hyperboloid panoramic vision system is shown in Figure 2.

The numerical relation among the three parameters *a, b, ξ* is simulated with MATLAB. From the calculation results (left of Figure 3), when *b* gets the minimum value and *a* gets the maximum value, the vertical field of view can reach the maximum value. From the contour, the influence of parameter *b* on the vertical field of view is far greater than that of parameter *a*. The parameters of GX2750 camera used in the laboratory are as follows: sensor model ICX694, resolution 2200 × 2750, pixel size 4.54 × 4.54 μm; the distance between the panoramic camera and the ground is 1.3 m.

The design of hyperboloid mirror parameters should also meet the following conditions:The upper boundary point of vertical field of viewMax *P_max_* and lower boundary point *P_min_*. The projection points of Min should be in the image plane.Since the angle of view above the horizontal plane of the hyperbolic reflector is basically reflected sky and belongs to the invalid area, the upper boundary angle of the vertical field of view angle is less than 0°.The viewpoint of the perspective camera should include the maximum reflection area of the hyperbolic mirror. Because the field of view angle of the perspective camera is inversely proportional to the focal length, it is necessary to select the lens with small focal length when selecting the focal length of the perspective camera. In the experiment, the focal length of the smallest one-inch lens is 16 mm, which is cheap and common.

The solution set of condition *a* and condition *b* is shown in Figure 3, and the solution set with condition restriction is shown in Figure 4. By solving the intersection of the two solutions shown in Figure 4, the values of hyperboloid mirror parameters *a* and *b* can be obtained. In order to minimize HOVS, as can be seen from Figure 5 that *a* = 48, *b* = 44, and its physical diagram is shown in Figure 5 (right).

2.Mirror mounting disk rotation module, mirror height adjustment module, and visual positioning module.

In order to facilitate the rapid switching of different hyperboloid mirrors to obtain different field of view and depth of field information, for this special experimental requirement, there is currently no special device for automatically switching mirrors, adjusting the vertical field of view, and adjusting the depth of field and image clarity for this special experimental requirement. The existing binocular stereoscopic panoramic vision imaging mirror switching technology mainly adopts manual methods. There are many defects in the way of manually switching mirrors, such as long switching time, low switching efficiency, poor switching accuracy, and so on. In the process of multiple switching of the reflector, it is easy to cause mirror contamination and mirror thread damage, which causes unnecessary resource loss.

Based on the above situation, the authors of this paper have designed a mirror mounting rotation module, a mirror height adjustment module, and the vision fixed module. The mirror mounting plate rotation module is used to mount four hyperboloid mirrors with different parameters and drive the mirror to rotate. The visual positioning module adjusts the coincidence degree of the central axis of the mirror and the optical axis of the fluoroscopic camera through PID control and the position of the label on the mirror mounting plate. The flow chart of the visual positioning module is shown in Figure 6. The reflector height adjustment push rod module adjusts the distance between the reflector and the main point of the camera to achieve a single point of view and large depth of field visual effect. The structure of monocular panoramic vision system and binocular stereo vision system are shown in Figure 7 and Figure 8. 

3.Principle of depth information acquisition in HOVS.

The epipolar geometry principle and formula derivation process of the horizontal binocular stereoscopic panoramic vision systems is very different from ordinary perspective imaging. The ordinary perspective camera is a small hole imaging model and the panoramic vision systems use a panoramic model. The ordinary perspective camera uses rectangular coordinate systems and panoramic vision systems use a cylindrical coordinate system. Therefore, the depth information of binocular stereo panoramic vision cannot be obtained by ordinary perspective cameras. In order to facilitate the calculation, the horizontal binocular stereo panoramic vision systems’ coordinate systems in Figure 9 are simplified as Figure 10. $O_{1}$ and $O_{2}$ is the upper focus of two hyperboloid mirrors, $P_{w}$ is the world point in the world coordinate system.

The horizontal distance between two single-view panoramic vision systems is **b**, and there is a world point $P_{w}$. In the panoramic vision systems, the world coordinates of camera_left and camera_right are $P_{w1}=\left(r_{w1};\varphi_{w1};z_{w1}\right),\;P_{w2}=\left(r_{w2};\varphi_{w2};z_{w2}\right)$. The projection points on the camera_left hyperboloid and camera_right hyperboloid are $P_{h1},P_{h2}$. The projection points on the right panoramic image plane and camera_left panoramic image plane are $P_{i1},P_{i2}$, $r_{p}$ is the distance between $P_{w}$ and $O_{1}O_{2}$, The intersection points of $P_{w}O_{1}$,$P_{w}O_{2}$ and left hyperboloid, right hyperboloid are $P_{h1},P_{h2}$.The plane Π is the plane of the *x*-axis of the rectangular coordinate systems and perpendicular to the *z*-axis. The projection of point $P_{w}$ on the Π plane is ${P_{w}}^{\prime }$. The intersection points of $P_{w}O_{1}$ and camera_left hyperboloid mirror is ${P_{h1}}^{\prime }$.The intersection points of $P_{w}O_{2}$ and camera_ right hyperboloid mirror is ${P_{h2}}^{\prime }$.The angle between ${P_{w}}^{\prime }{P_{h1}}^{\prime },{P_{w}}^{\prime }{P_{h2}}^{\prime }$ and X axis is ${\varphi_{1}}^{\prime },{\varphi_{2}}^{\prime }$. Because φ where is no change after the nonlinear transformation of hyperboloid mirror, and the number of projection points in panoramic image is small, then the value of $\varphi_{i}$ is equal to that of the environment point φ Value. And plane ${P_{w}}^{\prime }O_{1}P_{w}$ is perpendicular to the plane Π, so ${P_{w}}^{\prime }$, $P_{w}$ and $P_{i}$ are same value ($\varphi_{1}={\varphi_{1}}^{\prime }=\varphi_{i1};\varphi_{2}={\varphi_{2}}^{\prime }=\varphi_{i2}$).

In $\Delta {P_{w}}^{\prime }O_{1}O_{2}$, according to the sine theorem:(4)$$\frac{{P_{w}}^{\prime }O_{2}}{\sin{\varphi_{1}}^{\prime }}=\frac{{P_{w}}^{\prime }O_{1}}{\sin{\varphi_{2}}^{\prime }}=\frac{O_{1}O_{2}}{\sin{\varphi_{3}}^{\prime }}$$
(5)$${P_{w}}^{\prime }O_{1}=b\frac{\sin\left(\varphi_{i2}\right)}{\sin\left(\varphi_{i2}-\varphi_{i1}\right)}$$
(6)$${P_{w}}^{\prime }O_{2}=b\frac{\sin\left(\varphi_{i1}\right)}{\sin\left(\varphi_{i2}-\varphi_{i1}\right)}$$

In $\Delta {P_{w}}^{\prime }P_{w}O_{1}$, due to ${P_{w}}^{\prime }$ is the vertical projection point of $P_{w}$ on plane Π, so: (7)$$\tan\left({P_{w}}^{\prime }O_{1}P_{w}\right)\;=\frac{{P_{w}}^{\prime }P_{w}}{O_{1}{P_{w}}^{\prime }}\;$$

The back-projections is performed from the image plane pixel $P_{i}\left(x_{i},y_{i},z_{i}\right)$ to environment point $P_{w}\left(x_{w},y_{w},z_{w}\right)$. In this paper we discuss the transformation relations between the two points.
(8)$$k=\frac{b^{2}fc+ab^{2}\sqrt{x_{i}^{2}+y_{i}^{2}+f^{2}\;}}{a^{2}\left(x_{i}^{2}+y_{i}^{2}\right)-b^{2}f^{2}}$$

From the polar coordinate back-projection transformation and Equation (8), through the analysis and derivation of the polar coordinate equation, finally, the environmental point *P_w_* can be calculated from the wold coordinate in the left and right camera coordinate systems, it is called depth information.
(9)$$camera\_left:\;\left\{\begin{array}{c} r_{w1}=b\frac{\sin\left(\varphi_{i2}\right)}{\sin\left(\varphi_{i2}-\varphi_{i1}\right)}\\ \varphi_{w1}=\varphi_{i1}\\ z_{w1}=b\frac{\sin\left(\varphi_{i2}\right)}{\sin\left(\varphi_{i2}-\varphi_{i1}\right)}\times \frac{-2c-fK}{kr_{i1}} \end{array}\right.\;$$
(10)$$camera\_right:\left\{\begin{array}{c} r_{w2}=b\frac{\sin\left(\varphi_{i2}\right)}{\sin\left(\varphi_{i2}-\varphi_{i1}\right)}\\ \varphi_{w2}=\varphi_{i2}\\ z_{w2}=b\frac{\sin\left(\varphi_{i1}\right)}{\sin\left(\varphi_{i2}-\varphi_{i1}\right)}\times \frac{-2c-fK}{kr_{i2}} \end{array}\;\right.$$

### 3.2. HOVS Calibration

#### 3.2.1. Calibration Principle of Monocular Panoramic Vision Systems for Single-View Hyperboloid Mirror

Since the mathematical model and projection model of hyperboloid mirror panoramic vision systems are completely different from those of an ordinary perspective camera, a new calibration method is adopted, which first calibrates the hyperboloid mirror parameters (external parameters) and then calibrates the perspective camera parameters (internal parameters).

In order to calibrate the basic parameters of the hyperboloid mirror, the hyperboloid equation is used to model the calibration algorithm. In order to make the calibration algorithm more robust and extensive, the polynomial series is used to fit the shape of hyperboloid mirror.

The calibration principle of hyperboloid mirror panoramic vision systems is shown in Figure 11 above. The point *P_w_* on the chessboard calibration board changes nonlinearly to the point $P_{h}$ on the hyperboloid mirror, then through perspective transformation, it becomes the point $P_{i}$ in the image plane. Where the set of $P_{w}$ points is denoted by $N_{ij}$, the set of $P_{i}$ points is denoted as $n_{ij}$. From the above world point $P_{w}$, the transformation process shows that:(11)$$K_{ij}\cdot [\begin{array}{c} m_{ij}\\ f\left(m_{ij}\right) \end{array}]=[R_{0}^{1}|t_{0}^{1}]\cdot N_{ij}$$

In the process of $P_{w}\to P_{h}$, **R** is the rotation matrix of 3 × 3, t is the translation vector of 3 × 1, and $K_{ij}$ is the depth coefficient scalar, which is used to measure the world point coordinates $P_{w}$ in the same direction and the projection point coordinates of the reflector $\;P_{h}$. Thus, Equation (11) can be changed into Equation (12).

(12)$$K_{ij}\cdot \left[\begin{array}{c} u_{ij}-c_{x}\\ v_{ij}-c_{y}\\ f\left(u_{ij}-c_{x},v_{ij}-c_{y}\right) \end{array}\right]=\left[\begin{array}{cccc} r_{11} & r_{12} & r_{13} & t_{1}\\ r_{21} & r_{22} & r_{23} & t_{2}\\ r_{31} & r_{32} & r_{33} & t_{3} \end{array}\right]\cdot \left[\begin{array}{c} x_{ij}\\ y_{ij}\\ 0\\ 1 \end{array}\right]\;$$where (13)$$\;r^{\tau }{}^{2}=u^{\tau }{}^{2}+v^{\tau }{}^{2}$$(14)$$\;f\left(u_{ij}-c_{x},v_{ij}-c_{y}\right)\;=f\left(u^{\tau },v^{\tau }\right)\;=a_{0}+a_{1}r^{\tau }+a_{2}r^{\tau }{}^{2}+a_{3}r^{\tau }{}^{3}+\cdots +a_{N}r^{\tau }{}^{N}$$

$x_{ij},\;y_{ij}$ are the corner coordinates of the chessboard; $u_{ij},\;v_{ij}$ are the projection point coordinates of chessboard corner in pixel coordinate systems.

$c_{x},\;c_{y}$ are the coordinates of the center point of the image plane; $u^{\tau },\;v^{\tau }$ are the projection point coordinates of the projection point in the image plane coordinate systems.

By expanding Equation (12), it can be known that only three sets of chessboard angular coordinates are needed to solve the hyperboloid mirror external parameter matrix. However, in practical work, due to the influence of environmental noise and acquisition error, more corner coordinates are used to form the least squares problem.

By constructing reprojection residuals $S_{ij}$, the iterative calculation is carried out along the descending direction of the gradient until the residual result is less than the accuracy, and the iteration is stopped. At this time, the parameters corresponding to the optimal solution are the internal parameter matrix. The mathematical model of nonlinear optimization is as follows:(15)$$\arg\begin{array}{c} min\\ \left(R,t,a_{0},a_{1},\ldots \ldots \ldots a_{N}\right) \end{array}\overset{m}{\underset{i=1}{\sum }}\overset{n}{\underset{j=1}{\sum }}||\;S_{ij}\;|{|}_{2}^{2}$$
(16)$$S_{ij}=n_{ij}-n_{ij}^{\tau }$$
where

$n_{ij}$ is the observation value and $\;n_{ij}^{\tau }$ is the reprojection point value.

#### 3.2.2. Calibration Principle of Binocular Stereo Panoramic Vision Systems with Single-View Hyperboloid Mirror

When calculating the stereo depth of binocular stereo panoramic vision systems, the two monocular panoramic vision systems are arranged in strict accordance with the horizontal alignment. In fact, because of the installation accuracy and other reasons, the two panoramic vision systems are not strictly aligned horizontally, there is always a deviation, and the baseline **B** between the two monocular panoramic vision systems cannot be obtained by actual measurement. Therefore, before using the binocular stereo panoramic vision systems, it is necessary to calibrate it to determine its relative position relationship and baseline length. The calibration principle and structure are shown in Figure 12 and Figure 13.

The calibration of binocular stereo panoramic vision systems is easier than the calibration of the monocular panoramic vision system. The principle can refer to the ordinary binocular camera calibration method. After the monocular panoramic vision systems are calibrated, the accurate projection relationship can be determined, and various computer vision algorithms can also be used normally. By using the third-party vision detection library Aruco to complete the calibration of the binocular stereo panoramic vision systems, Aruco is an open-source augmented reality library that can be used to complete computer vision tasks such as tracking, recognition, and positioning. This research mainly uses Aruco’s GPS. It should be noted that the camera model supported by the Aruco library does not include a panoramic vision system, so the image detection part of the Aruco front end needs to be modified to expand the panoramic image to the hotspot area.

The coordinate systems relationship of binocular stereo panoramic vision systems calibration is shown in Figure 14. The two panoramic vision systems are camera_left and camera_right respectively, and the attitude transformation matrix from camera_right to camera_left is $T_{c2}^{c1}=\;\left[R_{c2}^{c1},t_{c1}^{c2}\right]$, The calibration plate is a plate printed with a specific Aruco pattern. Two panoramic stereo vision systems detect and recognize the same Aruco calibration board at the same time, and get $T_{b}^{c1}=\;\left[R_{b}^{c1},t_{b}^{c1}\right]$ and $T_{b}^{c2}=\;\left[R_{b}^{c2},t_{b}^{c2}\right]$.
(17)$$\left\{\begin{array}{c} T_{c2}^{c1}=T_{b}^{c1}\cdot T_{b}^{c2}\\ T_{c2}^{c1}=\left[\begin{array}{cc} R_{b}^{c1} & t_{b}^{c2}\\ 0 & 1 \end{array}\right]\cdot \left[\begin{array}{cc} R_{b}^{c2}{}^{T} & -R_{b}^{c2}{}^{T}\cdot t_{b}^{c2}\\ 0^{T} & 1 \end{array}\right] \end{array}\right.$$

Equation (17) is the calculation process of the attitude transformation matrix between two panoramic vision systems. Generally, the optimal $T_{c2}^{c1}$ is obtained by repeatedly calculating multiple pairs of binocular images.

$T_{c2}^{c1}$ is a pose transformation matrix, which cannot be added or subtracted, it needs to be calculated many times to get the optimal solution.

It is also necessary to construct a nonlinear optimization problem to solve. Order $P_{ij}^{c1}$ is the coordinate of the *j*-th corner in the *i*-th image in camera_left coordinate systems; $P_{ij}^{c2}$ is the coordinate in the camera_right coordinate systems. According to the transformation relation, there is Equation (18).
(18)$$\;T_{c2}^{c1}\cdot P_{ij}^{c1}=P_{ij}^{c2}$$

According to Equation (14), the optimization problem is constructed, such as Equation (19).
(19)$$argmin_{T_{c2}^{c1}}\overset{n}{\underset{i=1}{\sum }}\overset{k}{\underset{j=1}{\sum }}||T_{c2}^{c1}\cdot P_{ij}^{c1}-P_{ij}^{c2}||{}_{2}^{2}$$

In Equation (18), $P_{ij}^{c1}$ and $P_{ij}^{c2}$ are the coordinates of the corner points on the Arco calibration plate observed by the left and right cameras in their respective coordinate systems. In particular, in order to avoid the problem that the orthogonality constraint of rotation matrix may lead to optimization failure in the process of optimization, in the actual calibration process, the rotation part of the pose transformation matrix $T_{c1}^{c2}$ uses quaternion to participate in the optimization, which is converted into rotation matrix after the optimal result is obtained. The whole calibration algorithm flow is shown in Algorithm 1.


**Algorithm 1** Aruco estimates tag attitude.  Input: ***n*** pairs of left and right panoramic images
**if *i* < *n* then**
  Binocular stereo panoramic image distortion correction  Image preprocessing, extracting Aruco tag  **If** Aruco tag is found in both left and right images **then**  Extract the posture of Aruco tag, $\sum_{i=1}^{n}\sum_{j=1}^{k}||T_{c2}^{c1}\cdot P_{ij}^{c1}-P_{ij}^{c2}||{}_{2}^{2}$ is added to the nonlinear optimization equations
  **else *i* = *i* + 1**

  **end if**

**Else**
  Solving nonlinear optimization equations with MATLAB  **if** the average reprojection error is less than ±5 pixels **then**     Calibration successful    **Else**   Calibration failed
  **end if**

**end if**



### 3.3. HOVS Image Expansion

#### Panoramic Image Expansion Algorithm Based on VCAM

The panoramic image expansion algorithm is mainly divided into three parts:

The first part: As shown in Algorithm 2 below, the mathematical model of panoramic vision systems is established, and the two-dimensional image points on the panoramic image are transformed into three-dimensional world point cloud through the mathematical model of panoramic vision systems;

The second part: As shown in the following Algorithm 3, the virtual camera model is established, and the three-dimensional world point cloud is transformed into a two-dimensional virtual vision expansion image through the virtual camera model. Finally, the panoramic expansion image is obtained after image quadratic interpolation and mean optimization;

The third part: As shown in the following Algorithm 4, calculate the panorama and time, and initialize panoramic_view_extract program, circularly publish image_topic, subscribe to image_ topic, initialize OCAM and VCAM and load the mapping file, through the defined DISPLAY and ADJUST macro to adjust the camera parameters and pose, and calculate the time t of loading OCAM model **t****_****i** and the time t of foreground expansion **t****_****p**. Finally, save the result in remap_view_ exact. Panoramic image expansion flow chart is shown in Figure 15.

The panoramic expansion algorithm includes Algorithms 2–4. The HOVS algorithm is to expand two same monocular panoramic images synchronously to form a panoramic image around the car body 360° 2D panoramic expansion within the scope.
**Algorithm 2** Transform 2D panoramic image to 3D world point cloud.  Input: 2D panoramic image  Output: 3D world point cloud  1: Start of OCAM algorithm  2: Loading OCAM model  3: Initialize OCAM  4: Read initial 2D panoramic image  5: Using polyval () to calculate Z coordinate of 3D world point  6: Using cam2word () to transform 2D image points into 3D world point clouds  7: Save world point cloud image  8: End of OCAM algorithm


**Algorithm 3** Image transformation from 3D world point cloud to panoramic expansion.  Input: 3D world point cloud  Output: Panoramic expansion image  1: VCAM algorithm starts  2: Loading VCAM model  3: Setting VCAM parameters  4: Loading 3D point cloud image  5: Update VCAM camera parameters  6: Using MD5 algorithm to verify the integrity of image expansion data transmission  7: Read image mapping file  8: Transformation from 3D point cloud image to 2D image points  9: Calculation of expanded image by bilinear interpolation algorithm  10: Real-time display of expansion image progress with progress bar  11: Mean optimization re_ MAP1 and re_ MAP2 image  12: Save the optimized image  13: End of VCAM algorithm



**Algorithm 4** Panoramic_view_extract.  Input: Panoramic theme, OCAM and VCAM  Output: Panoramic expansion time and panoramic expansion map  1: panoramic_ view_ Start of extract algorithm  2: Initialize panoramic_ view_ extract  3: Circularly publish image_ topic, image_ pub_ topic, camera_ model, virtual_ camera, point cloud_ topic, remap_ save_ path  4: Subscribe to image_ Topic, publishing to point_ Cloud and point_ Topic of cloud2  5: Initialize OCAM and VCAM, load mapping file  6: Through the defined display and adjust macro, create image_ Remap and control_ Panel window (parameters and pose of camera can be adjusted)  7: Load OCAM model into CV: mat map, and then map CV: mat map to VCAM: point_ Map calculates its time t_ i. Finally, VCAM: point_ The map is expanded quickly and the time t is calculated_ p  8: Finally, save the expanded map to remap_ save_ File  9: panoramic_ view_ End of extract algorithm


## 4. Experiments and Results

### 4.1. Calibration Experiment and Experimental Results of Monocular Panoramic Vision Systems with Single-View Hyperboloid Mirror

#### 4.1.1. Calibration Experiment

The 9 × 7 checkerboard is prepared, in order to avoid the influence of reflection of aluminum substrate around the checkerboard, the inner 7 × 5 checkerboard calibration panel is selected and aluminum alloy is used as the base plate. The parameters to be marked are $\;R,t,a_{0},a_{1},\ldots ,a_{N}$. The calibration procedure is as follows:

The first step is to collect nine high-quality panoramas, as shown in Figure 16 below.

The second step is to extract corner points on each panorama, as shown in Figure 17 below.

The third step is to set $R_{\epsilon }=I,{\;t}_{\varepsilon }=0$, the pose of the checkerboard calibration board in the camera coordinate systems in all panoramic images is shown in Figure 18 below.

In the fourth step, N = 2 is set. Through the result of the third step, the pose of the calibration plate, the pixel coordinates of the corners, and the R to be estimated can be obtained ${\;R}_{\epsilon }=I,{\;t}_{\varepsilon }=0$, the error equation of reprojection is constructed.

In the fifth step, nonlinear optimization and robust nonlinear optimization are used to solve the reprojection equation iteratively. The reprojection error of the optimized result is less than ±5 pixels, and the calibration result is finished and output. If it is larger, the fourth step is returned, *N* = *N* + 1.

The reprojection errors in the calibration results before optimization are as follows:Average reprojection error computed for each chessboard (pixels):

5.12 ± 4.02; 4.37 ± 2.97; 4.94 ± 3.82; 3.64 ± 2.95; 3.46 ± 2.83; 3.37 ± 2.78; 3.49 ± 2.79; 4.93 ± 3.94; 3.67 ± 2.85

Average error [pixels]; 4.108346 Sum of squared errors: 11985.065455

ss = −315.37231721852 0 − 0.00000141751 0.000000017656 − 0.00000000001

The results of nonlinear optimization are as follows:

ss = −215.68454874302 0 − 0.00002217486 0.00000006384 − 0.00000000004

Root mean square [pixel]: 2.315967

The results of robust nonlinear optimization are as follows

According to the average error of each image in Table 1, when *n* = 1, 2, 3, 4, and 5 are recorded in Table 2, the sum of squares of total reprojection error and average reprojection error can be calculated. It can be seen from the above table that when *n*= 1, f () becomes a linear function, and the fitting effect of single-view bisurface is very poor, resulting in large reprojection error. When *n* = 5, the optimizer cannot get the optimal result and the calibration fails. When *n* = 4, the reprojection error of calibration is the smallest, and the reprojection diagram of calibration plate corner is shown in Figure 19 below.

#### 4.1.2. Calibration Result

As shown in Figure 19 and Figure 20, the checkerboard can be correctly projected into the image, and the average reprojection error is about 3 pixels, so the calibration results can be considered to be effective. The calibration results of monocular vision systems of single-view hyperboloid mirror are shown in Figure 21 above, and the final results of calibration parameters are as follows:(20)$$N=4,\;f\left(r\right)\;=-219.7939+2.1487771\times 10^{-3}r^{2}-6.348362\times 10^{-6}r^{3}+4.161875\times 10^{-9}r^{4}$$

The coordinates of the central pixel of the mirror:(21)$$\;\left(X_{c},Y_{c}\right)\;=\;\left(993.103303,\;1337.177544\right).$$

Mirror position deviation:(22)$$\begin{array}{c} R_{\varepsilon }\;=\;\left(0.996229,-0.013996,-0.012452\right);\\ t_{\varepsilon }=\;\left(993.103303,\;1337.177544,\;0.0\right) \end{array}$$

After calibration, the coordinate of the rotation center of the mirror in the image plane is represented by a red circle, which is close to the image center of the mirror. The rotation vector of the position deviation of the mirror is expressed as $R_{\varepsilon }=\;\left(0.996229,-0.013996,-0.012452\right)$ (the first term is close to 1, the second and third terms are almost close to 0, which indicates that the error between the mirror and the panoramic vision systems is very small. 

### 4.2. Calibration Experiment and Experimental Results of Binocular Stereo Panoramic Vision Systems with Single-View Hyperboloid Mirror

After the monocular panoramic vision systems is calibrated, the horizontal binocular stereo panoramic vision systems can be calibrated. Firstly, the calibration environment is built, and the equipment used is as follows.

#### 4.2.1. Calibration Experiment

After the monocular panoramic vision systems is calibrated, the horizontal binocular stereo panoramic vision systems can be calibrated. Firstly, the calibration environment is built, and the equipment used is as follows: Two monocular panoramic vision systems, as shown in Figure 22.

4.One Aruco calibration board, as shown in Figure 23.

Two monocular panoramic vision systems are pre-calibrated and aligned on the same base. The size of the calibration plate for the Aruco logo is 1200 × 1200 mm, the pattern of the Aruco marker map is used as an Aruco tag to improve the stability of the Aruco detection algorithm.

The specific calibration steps are as follows:Collect the images of two panoramic vision systems at the same time, and keep the images containing the Aruco calibration board as far as possible. A total of 12 pairs of images are collected.According to the calibration results of the panoramic vision systems and the back-projection transformation, the collected image is partially expanded and the distortion is corrected. Figure 24 shows the partially expanded image of the panoramic vision systems after the back-projection transformation.

3.The position and posture of the Aruco calibration plate in each image are obtained by using the Aruco detection algorithm (see Figure 25 and Figure 26).

4.According to the position and pose of the Aruco calibration board in each image, the coordinates P of each corner on the calibration board in the current camera coordinate systems are calculated $P_{ij}^{c1}$ and $P_{ij}^{c2}$.5.According to Equation (23), the nonlinear optimization problem is constructed and solved by MATLAB.6.Calculate the reprojection error, and the calibration is considered successful when the average is within 10 pixels.

After MATLAB calculation, the final calibration results are as follows:Equation (23) is the rotation matrix and translation vector between two panoramic vision systems.The baseline length between the two panoramic vision systems is 0.64 m, the deviation in Y direction is −0.007 m, and the deviation in Z direction is −0.014 m.
(23)$$T_{c2}^{c1}=\left[\begin{array}{cccc} 0.999 & -0.020 & -0.012 & -0.641\\ 0.020 & 0.999 & 0.015 & -0.006\\ 0.011 & -0.016 & 0.999 & -0.013\\ 0 & 0 & 0 & 1 \end{array}\right]$$

It can be seen that the diagonal value of the rotation matrix in Equation (23) is close to 1, which indicates that the attitude relationship between the two panoramic vision systems is basically aligned, as shown in Figure 27.

#### 4.2.2. Calibration Result

Finally, the reprojection error of 64 pairs of images is shown in Figure 28. Camera index represents the serial number of the image pair, a total of 64 pairs. The left figure of Figure 28 shows the projection position distribution of 64 pairs of images, and the coordinate unit is pixel. The right figure of Figure 28 is the superposition of the average reprojection error of 64 pairs of images. It can be seen that the Y direction of the reprojection error of each pair of 64 pairs of images is in the same direction ± within 5 pixels and most of the X direction is in the ± within 5 pixels; thus, the calibration result is considered valid.

### 4.3. Experiments and Results of Image Expansion in HOVS

Experimental setting: Xishan campus of Beijing University of technology.Experimental equipment: Unmanned vehicle (Modified by BAIC EC180) and perception platform, HOVS systems (GX2750 camera resolution 2700 × 2200), computer, and other necessary equipment. Unmanned systems experimental platform as shown in Figure 29.Experimental evaluation index: The new HOVS systems can expand the panoramic image in real time with less distortion. As shown in Figure 30 below.Experimental method: By driving the vehicle in Xishan campus, the collected panoramic image is expanded in real time with small distortion.

Panorama resolution: At 2700 × 2200 for single panoramic vision systems.Resolution of unfolded image: Three directions of single panoramic vision systems, each direction 1200 × 700 (secondary interpolation), a total of six directions of unfolded image.Algorithm efficiency: Using the 2.3 Ghz main frequency processor, multithreading parallel processing, six direction pictures are expanded at the same time, each frame averages 4 to 5 ms (panoramic video frame rate is 10 fps), which can achieve the effect of real-time panoramic expansion.Algorithm effect: Using the mathematical model of panoramic calibration to interpolate the missing pixels, the image distortion after interpolation is significantly reduced (does not affect the typical feature extraction). Effect of partial expansion is as shown in Figure 31.

It can be seen from the local panoramic expansion effect picture in the above figure that the interpolation algorithm based on the mathematical model of panoramic vision systems can still maintain the texture features and lighting conditions of the environment at a higher resolution in the edge position, with large distortion of the panoramic image.

## 5. Conclusions and Future Work

In this paper, using the 360° imaging characteristics of the single-view hyperbolic catadioptric panoramic vision systems, two single-view hyperbolic catadioptric panoramic vision systems are symmetrically placed on the unmanned systems perception platform to construct a new type of binocular stereo panoramic perception. This system is particularly suitable for 360° obstacle real-time sensing and detection of military unmanned systems. With this requirement as a starting point, the imaging principle of the single-view hyperbolic catadioptric panoramic vision systems, the design of the hyperbolic catadioptric mirror, the mirror mounting disc rotation module, the mirror height adjustment module, and the vision positioning module are studied. The HOVS depth information acquisition principle, single-view hyperboloid mirror monocular panoramic vision systems calibration principle, single-view hyperbolic mirror binocular stereo panoramic vision systems calibration principle, and panoramic image expansion algorithm are based on VCAM.

The HOVS systems proposed in this paper can automatically switch mirrors and auto-focus to obtain scenes in different depths of field and different vertical angles of view. Aiming at the problem of panoramic camera calibration, a non-linear optimization method combined with the polynomial fitting method of the mirror is proposed to complete the calibration. For the calibration problem of the binocular stereo panoramic camera, the Aruco vision library is used to simplify the calibration process and improve the robustness and precision. At the edge of the panoramic image with larger distortion, the interpolation algorithm based on the VCAM mathematical model interpolates the missing pixels. The interpolated image can still maintain the texture characteristics and lighting conditions of the environment at a higher resolution. The panorama expansion algorithm can expand the picture in six directions at the same time, and each frame averages 4 to 5 ms (the panoramic video shooting frame rate is 10 fps), which can fully achieve the real-time panorama expansion effect.

Our future work will mainly focus on the integration, miniaturization, and generalization of the system. After the optimization of the system, the authors will also focus on panoramic vision slam and panoramic obstacle dynamic monitoring.

## Figures and Tables

**Figure 1 sensors-21-04708-f001:**
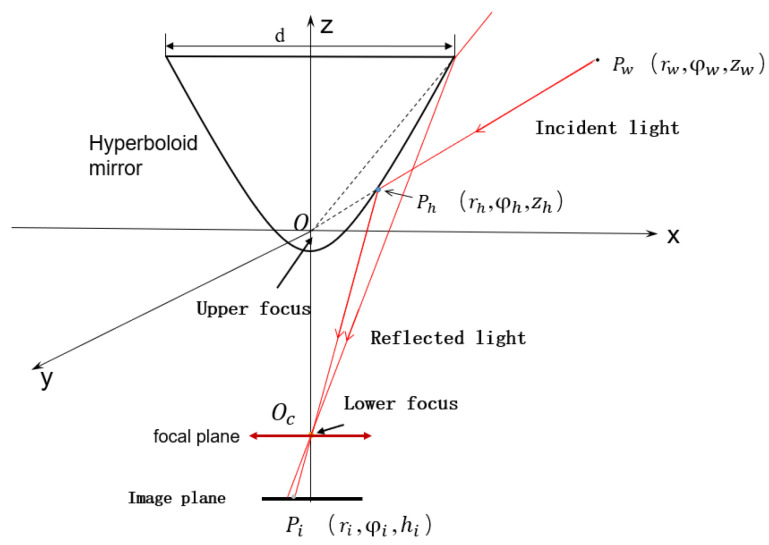
Cylindrical coordinate systems and imaging optical path of hyperboloid panoramic vision systems.

**Figure 2 sensors-21-04708-f002:**
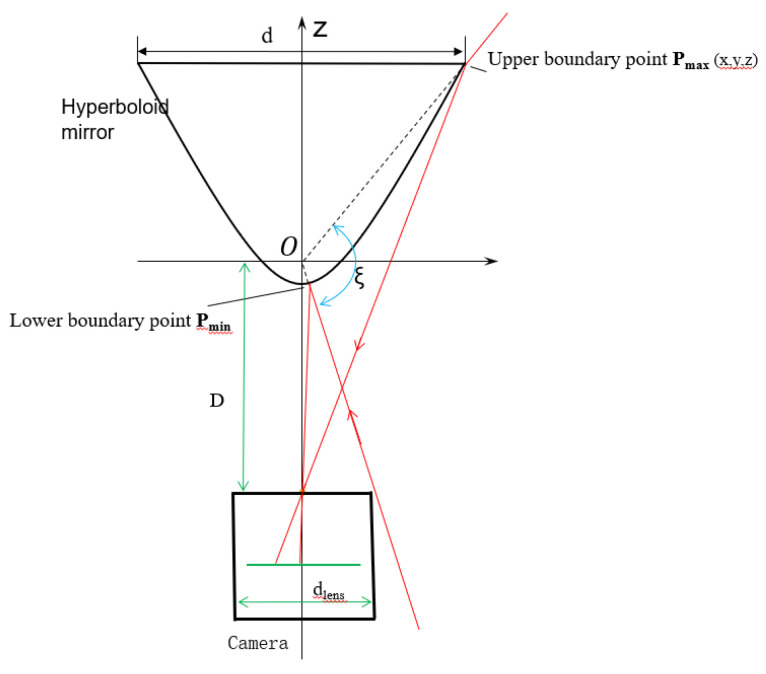
Schematic diagram of vertical field angle of view of hyperboloid panoramic vision systems.

**Figure 3 sensors-21-04708-f003:**
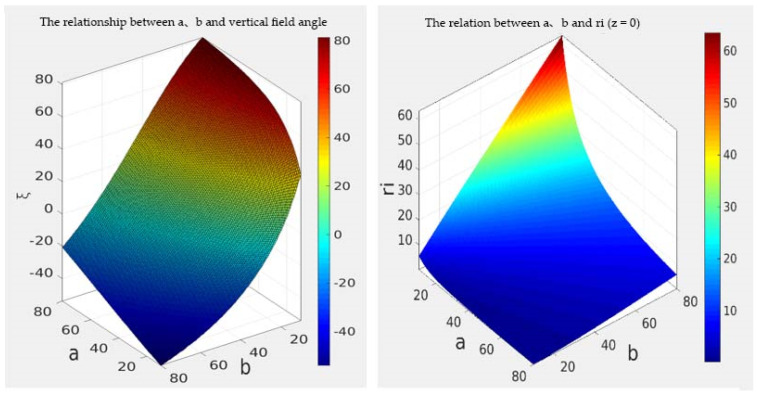
*a, b* and vertical field angle *ξ* relationship (**left**) and the relationship between *a, b* and *ri* vertical field angle (**right**).

**Figure 4 sensors-21-04708-f004:**
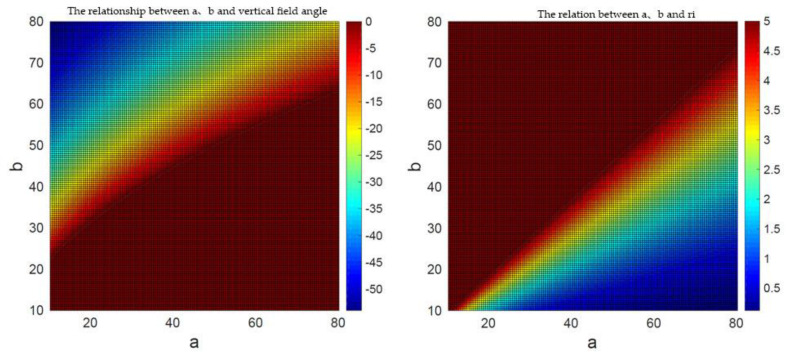
Feasible solutions of *a, b* and vertical field of view angle with constraints (**left**) and feasible solutions of *a, b* and *ri* with constraints (**right**).

**Figure 5 sensors-21-04708-f005:**
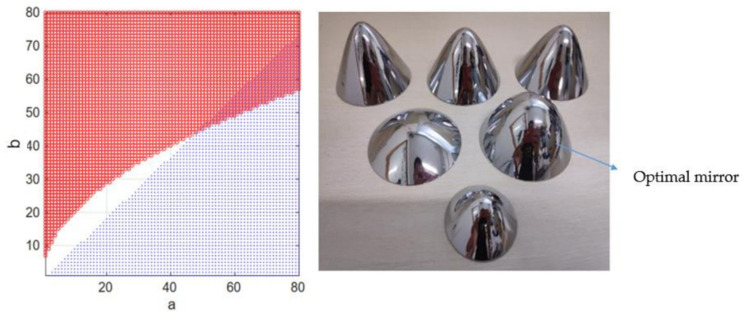
Feasible intersection of two solution spaces (**left**) and real image of mirror (**right**).

**Figure 6 sensors-21-04708-f006:**
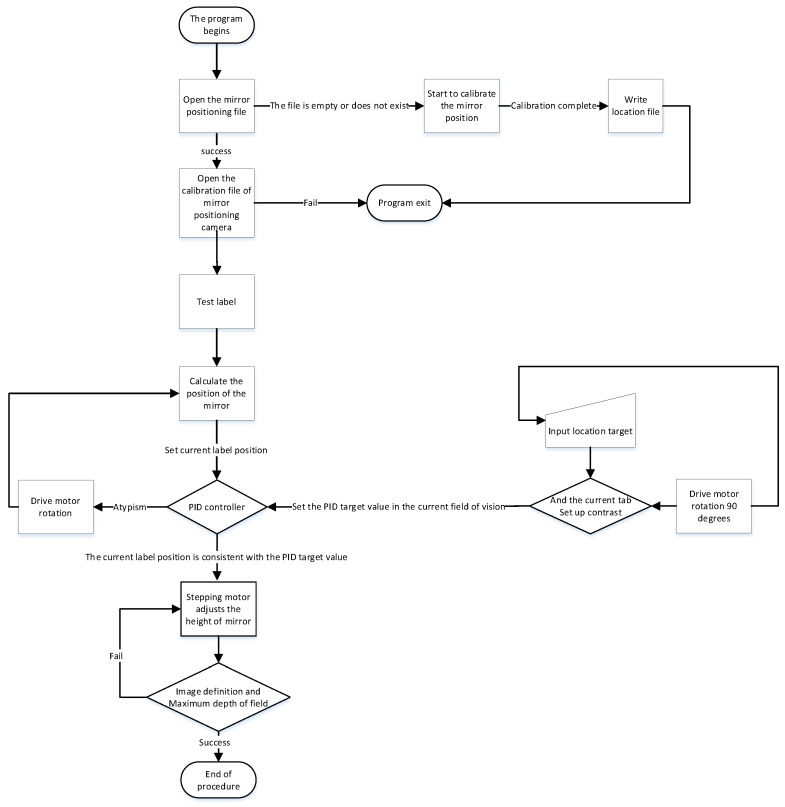
Feasible flow chart of visual positioning module.

**Figure 7 sensors-21-04708-f007:**
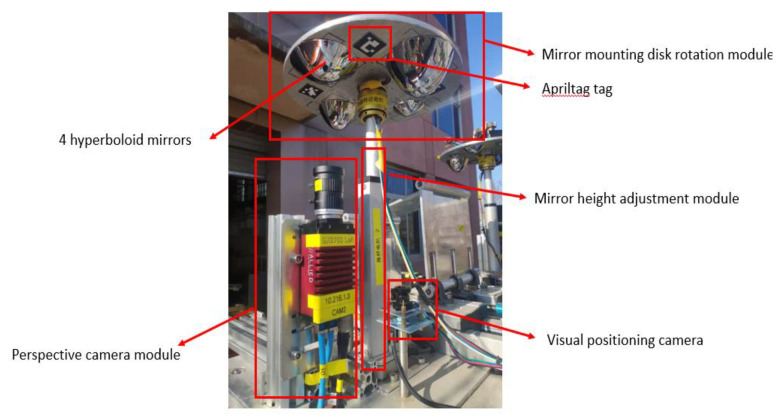
The physical picture of monocular panoramic vision systems.

**Figure 8 sensors-21-04708-f008:**
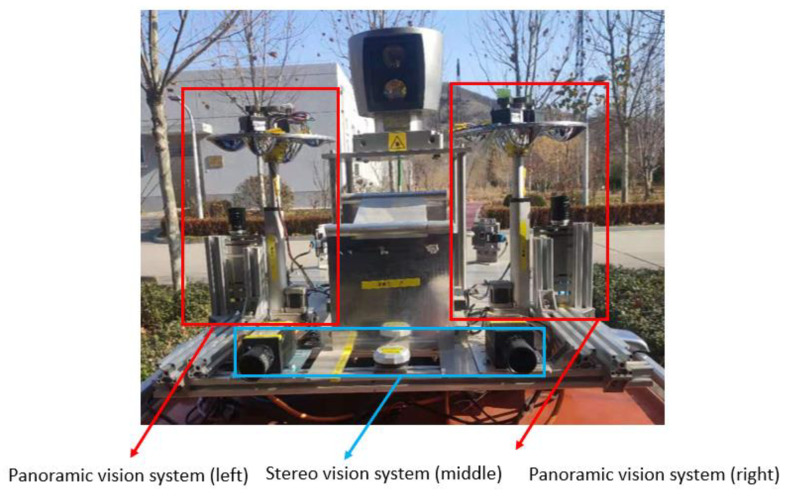
Physical map of binocular stereo panoramic vision systems.

**Figure 9 sensors-21-04708-f009:**
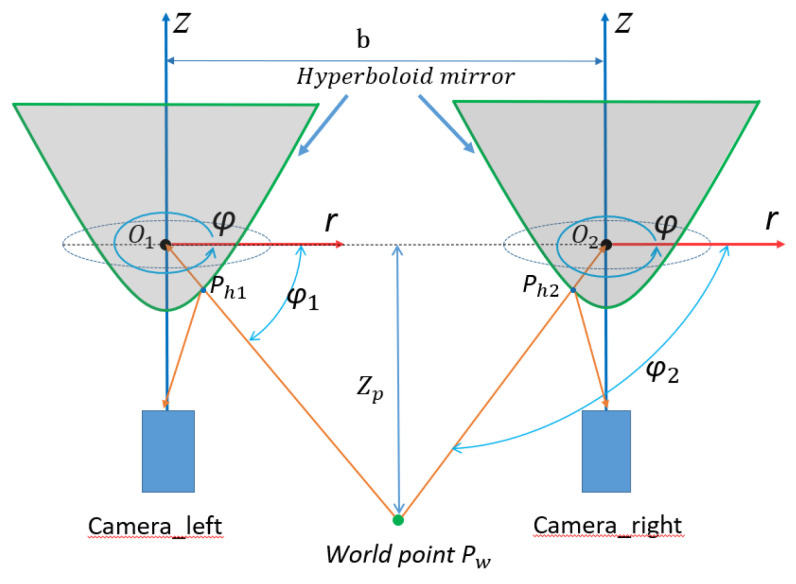
Coordinate systems of horizontal binocular stereo panoramic vision systems.

**Figure 10 sensors-21-04708-f010:**
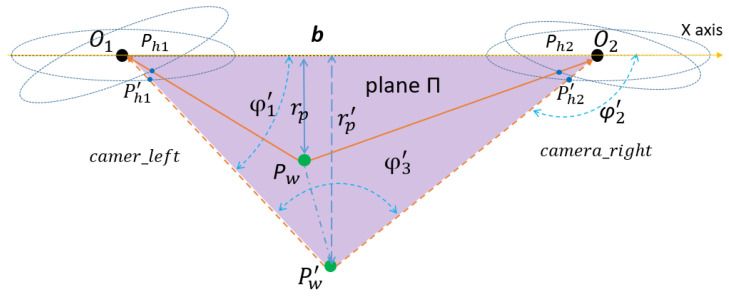
Geometric simplification of binocular stereo panoramic vision.

**Figure 11 sensors-21-04708-f011:**
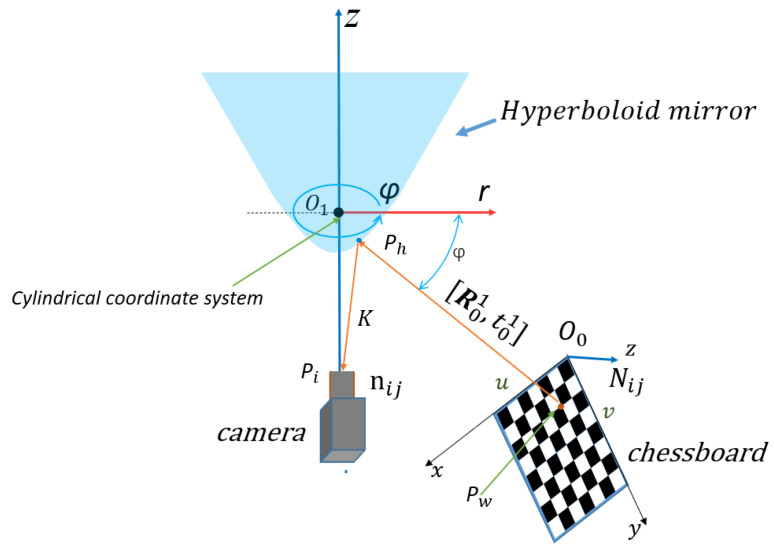
Calibration principle of panoramic vision systems for hyperboloid reflector.

**Figure 12 sensors-21-04708-f012:**
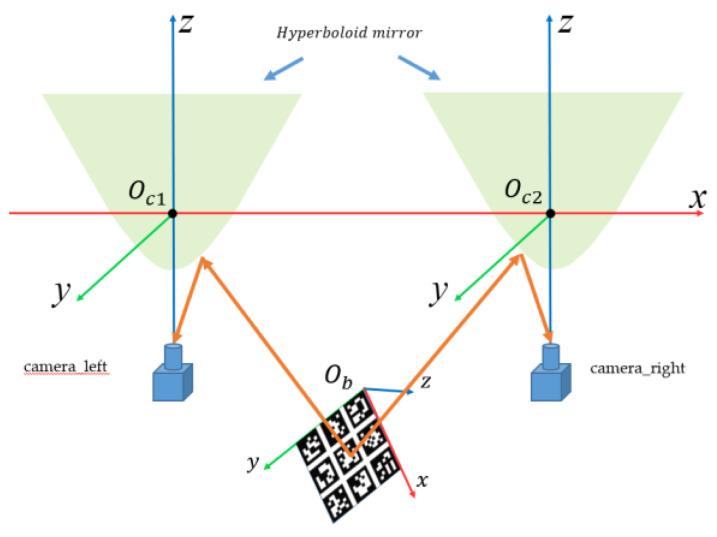
Schematic diagram of binocular stereo panoramic calibration.

**Figure 13 sensors-21-04708-f013:**
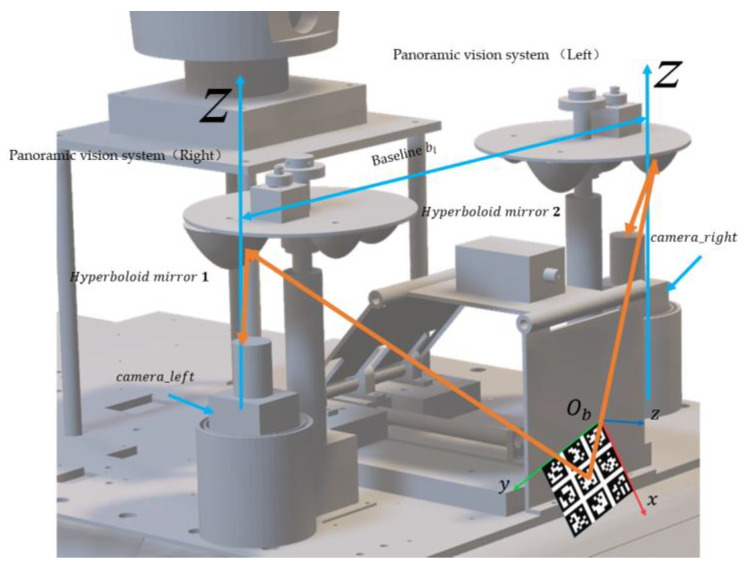
Structure diagram of binocular stereo panoramic calibration.

**Figure 14 sensors-21-04708-f014:**
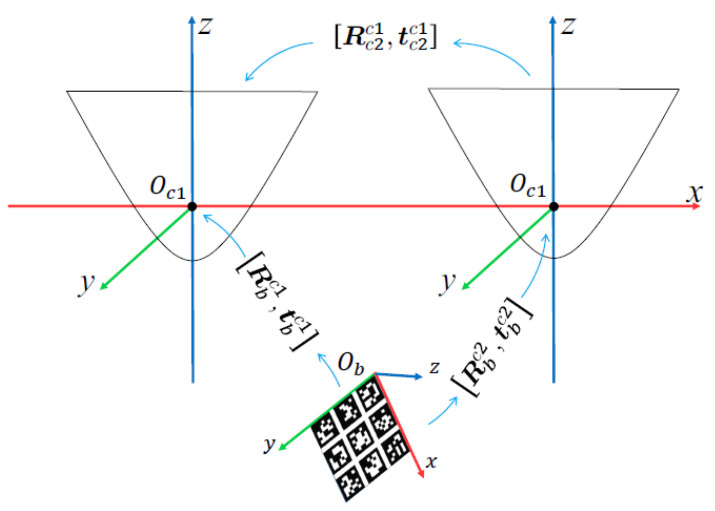
Coordinate systems relationship of binocular stereo panoramic calibration.

**Figure 15 sensors-21-04708-f015:**
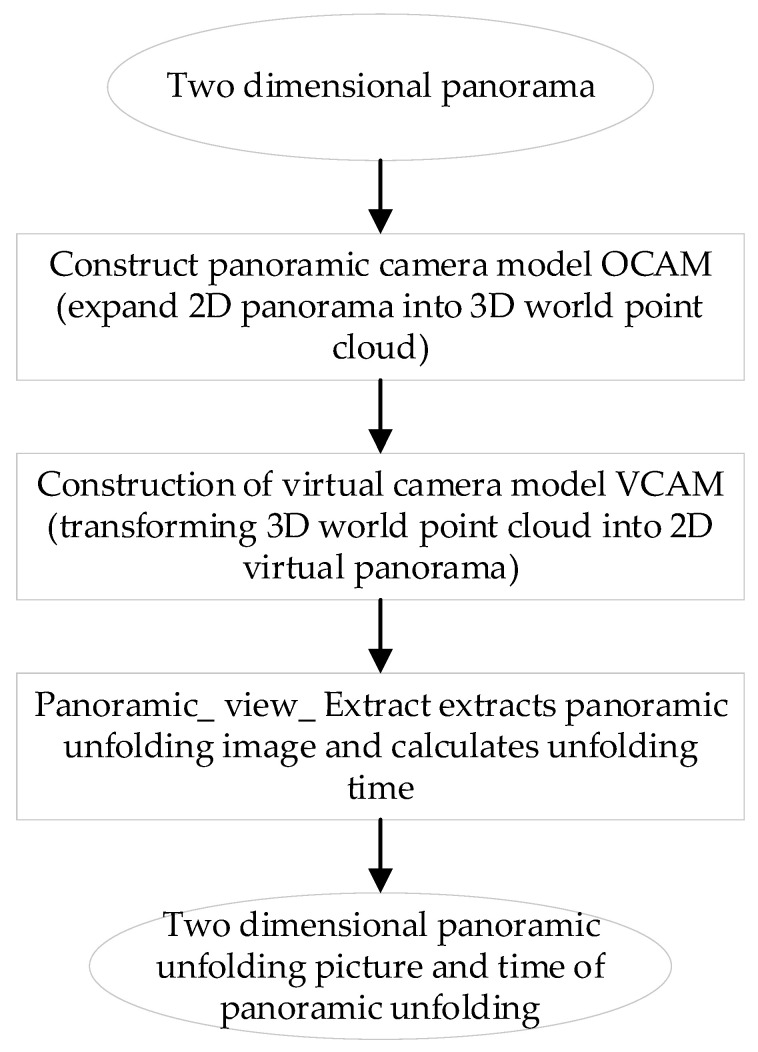
Panoramic image expansion flow chart.

**Figure 16 sensors-21-04708-f016:**
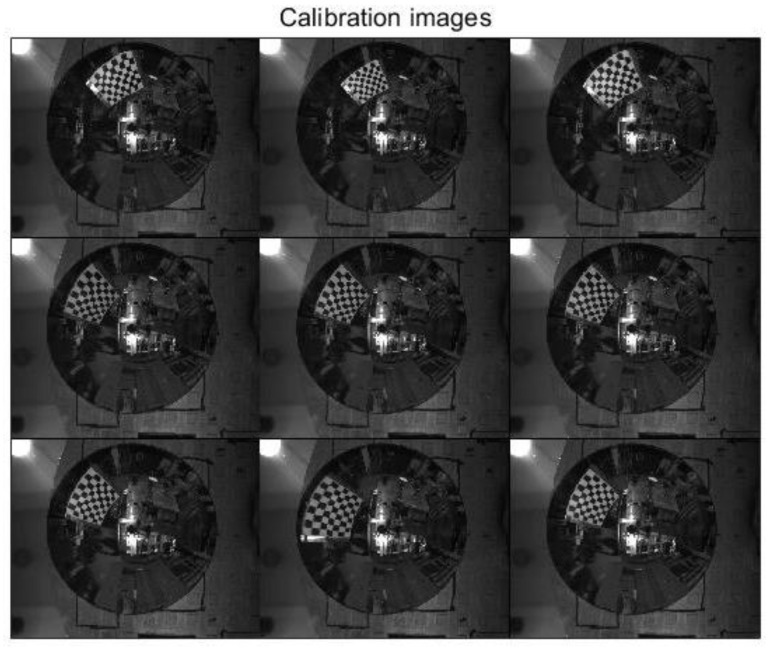
Read picture names.

**Figure 17 sensors-21-04708-f017:**
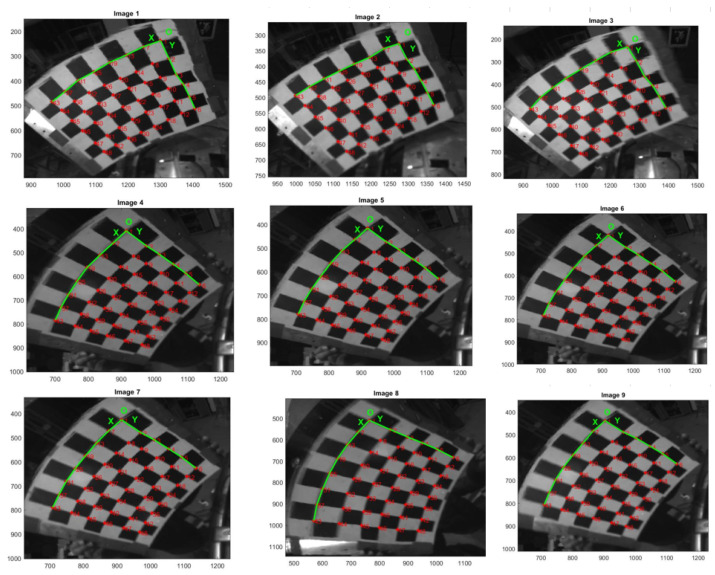
Extract grid corners.

**Figure 18 sensors-21-04708-f018:**
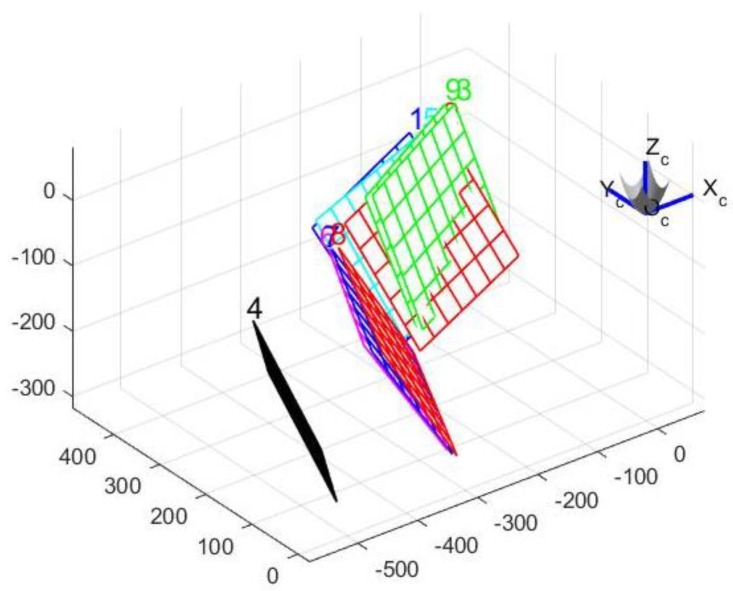
Show extrinsic.

**Figure 19 sensors-21-04708-f019:**
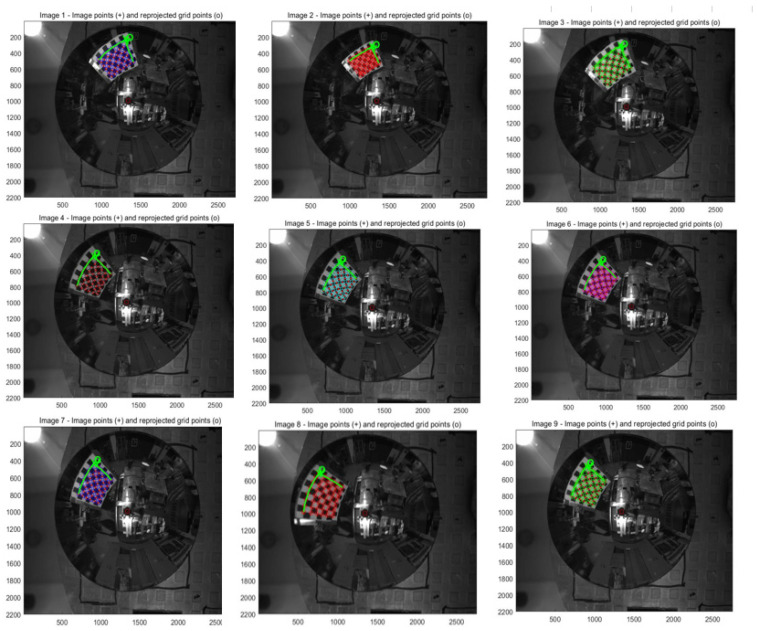
Reproject on images.

**Figure 20 sensors-21-04708-f020:**
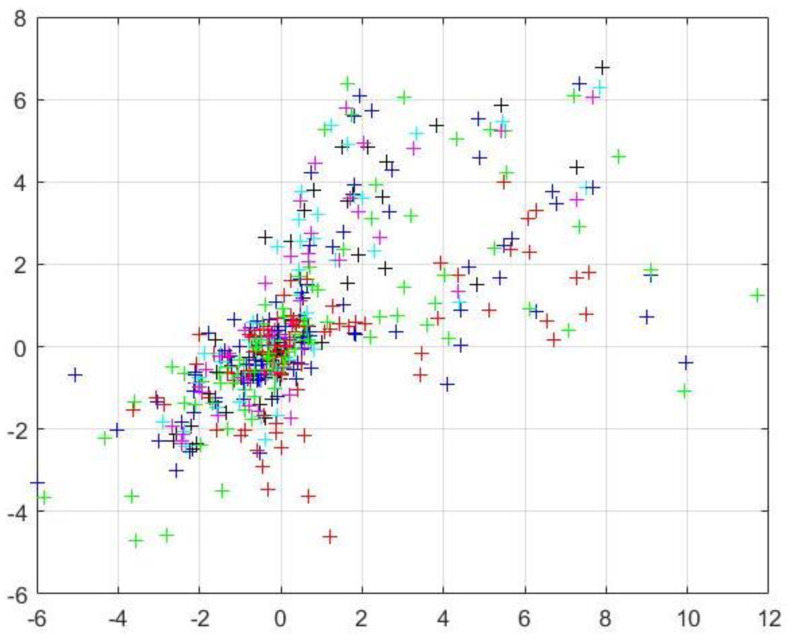
Reprojection error map.

**Figure 21 sensors-21-04708-f021:**
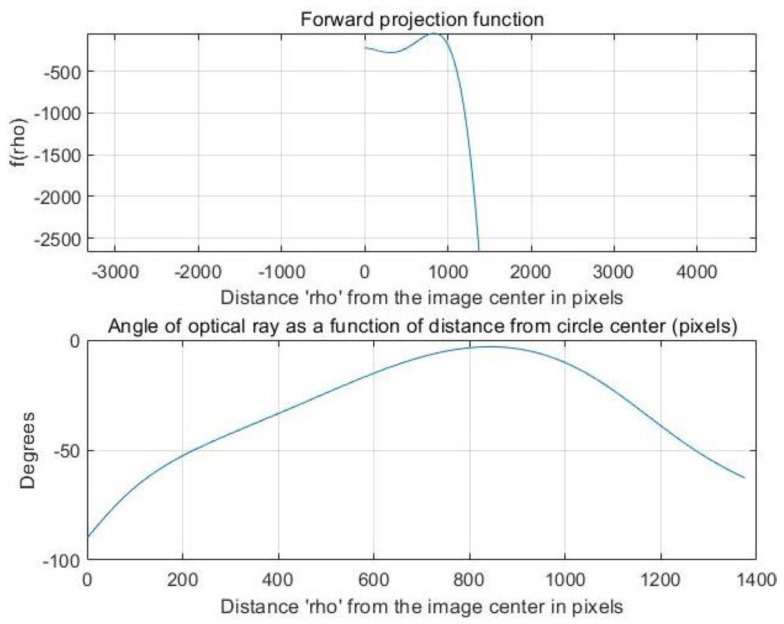
Calibration result chart.

**Figure 22 sensors-21-04708-f022:**
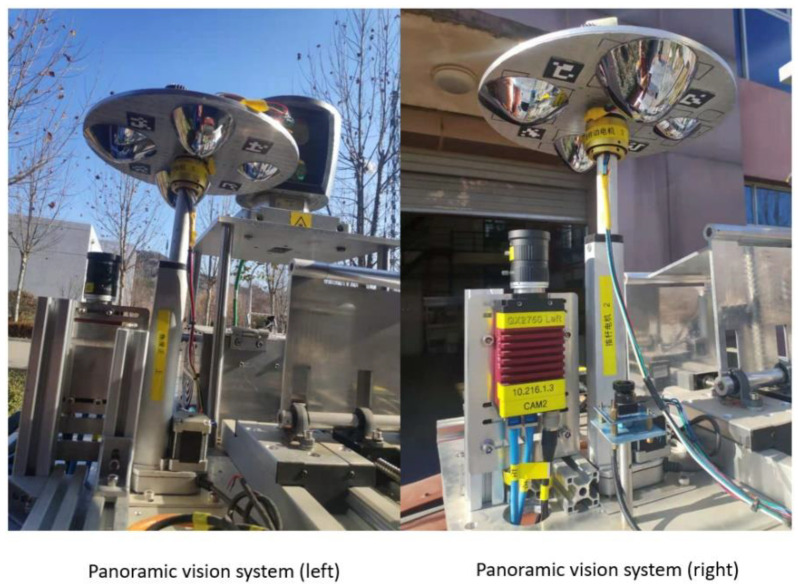
Binocular stereo panoramic systems to be calibrated.

**Figure 23 sensors-21-04708-f023:**
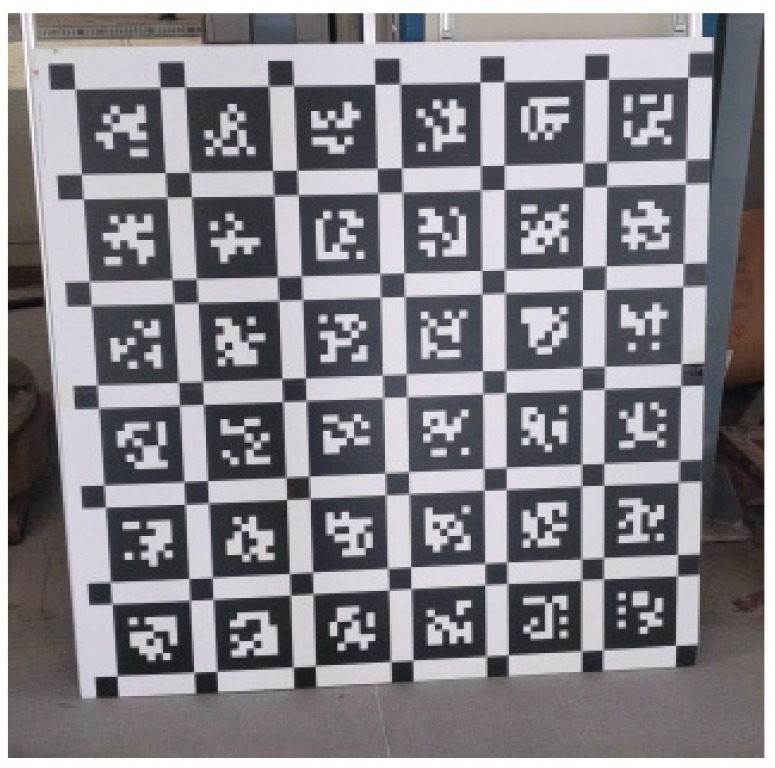
Aruco calibration plate.

**Figure 24 sensors-21-04708-f024:**
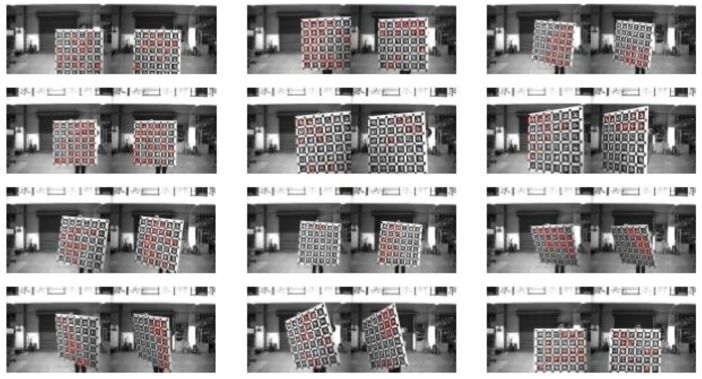
Twelve pairs of images collected at the same time (capture).

**Figure 25 sensors-21-04708-f025:**
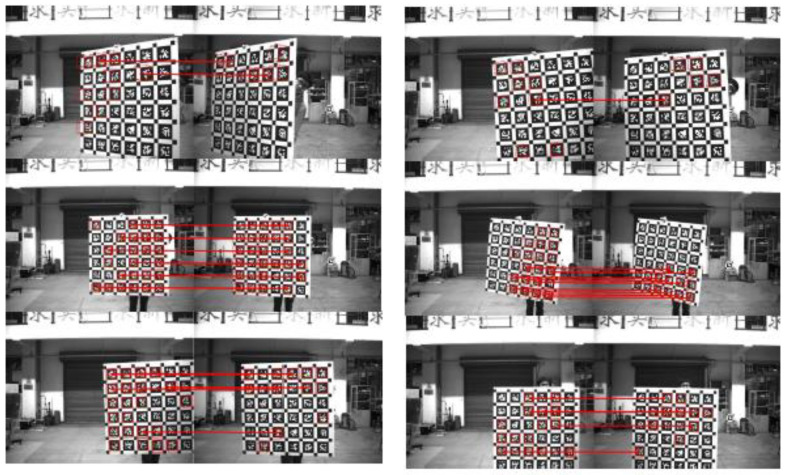
Extracted matching points and pose (1–6).

**Figure 26 sensors-21-04708-f026:**
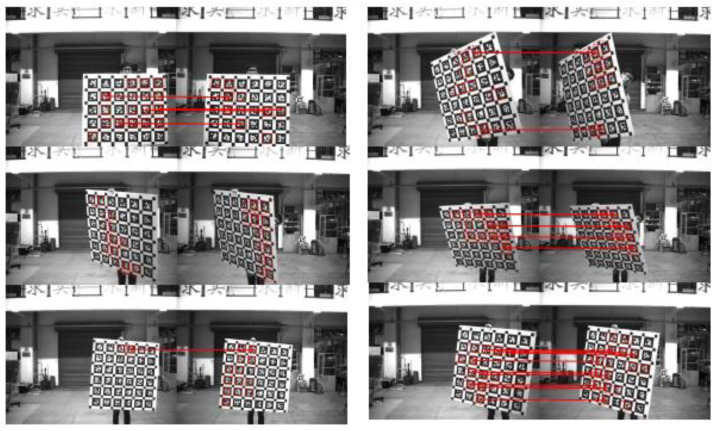
Extracted matching points and pose (7–12).

**Figure 27 sensors-21-04708-f027:**
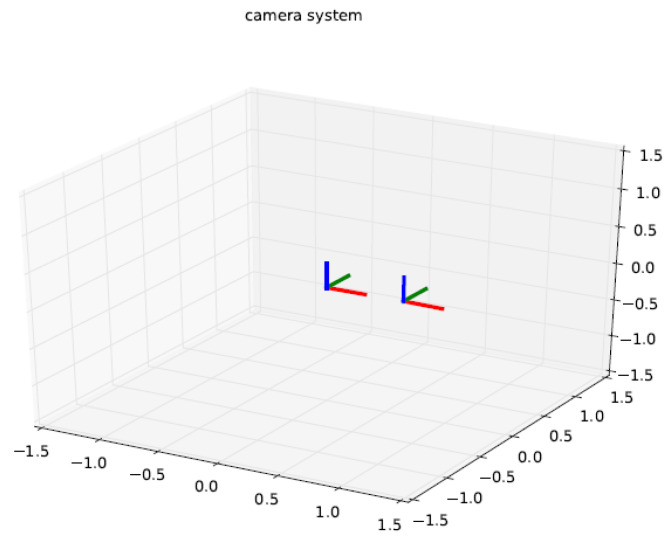
Spatial relationship of two panoramic vision systems.

**Figure 28 sensors-21-04708-f028:**
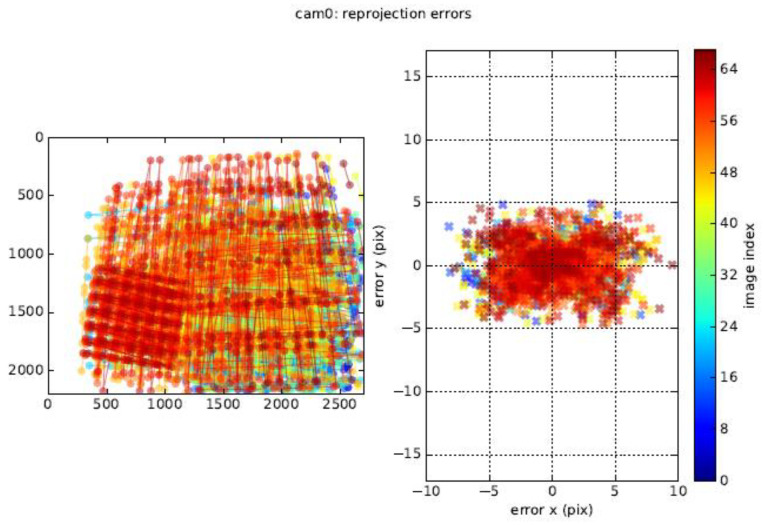
Reprojection error.

**Figure 29 sensors-21-04708-f029:**
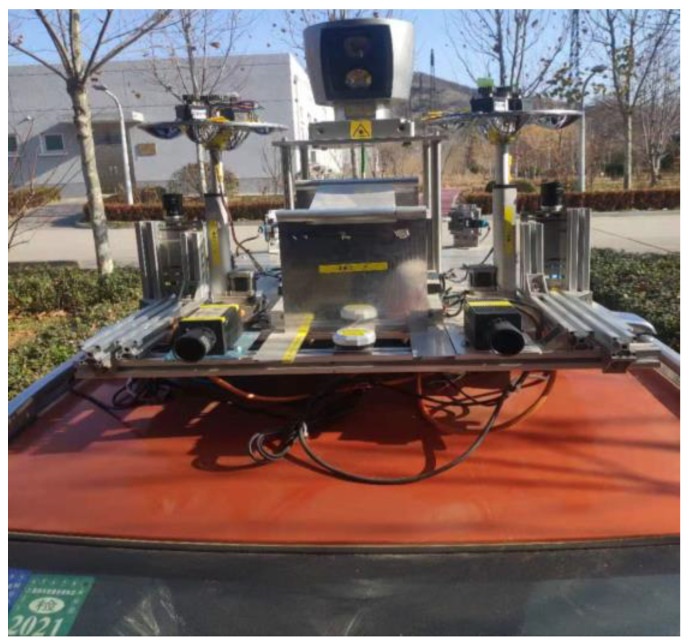
Unmanned systems experimental platform.

**Figure 30 sensors-21-04708-f030:**
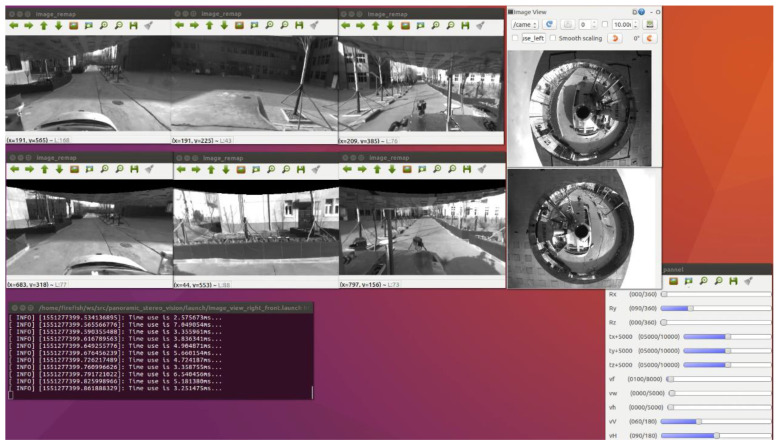
Six-direction real-time deployment rendering.

**Figure 31 sensors-21-04708-f031:**
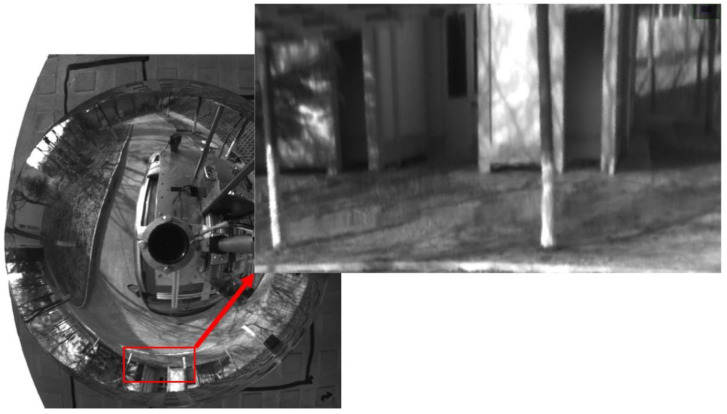
Effect of partial expansion.

**Table 1 sensors-21-04708-t001:** Average reprojection error table of optimized images when *n* = 1, 2, 3, 4, and 5.

Number	1	2	3	4	5
1	11.14 ± 08.03	4.43 ± 4.12	4.45 ± 4.48	2.92 ± 2.83	686.25 ± 337.40
2	12.59 ± 09.20	3.57 ± 3.06	3.26 ± 2.75	2.65 ± 2.47	812.85 ± 193.04
3	16.80 ± 09.92	4.47 ± 4.14	4.57 ± 4.38	3.05 ± 2.98	692.68 ± 322.51
4	11.72 ± 07.74	3.70 ± 3.52	3.51 ± 3.70	2.70 ± 2.29	615.23 ± 150.67
5	09.27 ± 06.73	3.44 ± 3.18	3.28 ± 3.46	2.61 ± 2.16	618.64 ± 147.67
6	08.52 ± 06.33	3.31 ± 3.21	3.13 ± 3.23	2.54 ± 2.17	618.98 ± 147.22
7	08.19 ± 06.45	3.19 ± 3.02	3.04 ± 3.05	2.58 ± 2.30	613.62 ± 148.42
8	17.66 ± 13.75	4.89 ± 3.40	5.26 ± 3.49	1.46 ± 1.32	NaN ± Nan
9	09.10 ± 06.80	3.19 ± 2.74	3.02 ± 2.81	2.69 ± 2.49	602.15 ± 150.47

**Table 2 sensors-21-04708-t002:** Overall reprojection error table of all images to be marked when *n* = 1, 2, 3, 4, and 5.

*n*	Sum of Total Weight Projection Error Square	Average Reprojection Error
1	94955.682	11.666
2	11290.089	3.798
3	11526.951	3.72
4	5340.762	2.57
5	NaN	NaN

## Data Availability

Not applicable.

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
