# Peer review of "Research on Design, Calibration and Real-Time Image Expansion Technology of Unmanned System Variable-Scale Panoramic Vision System"

_sensors, 2021, doi:10.3390/s21144708_

Round 1

Reviewer 1 Report

This paper presents an approach for panoramic view systems to be used in unmanned systems such as UAVs. The paper presents a really solid work and I consider that in conditions to be accepted. I just have some minor suggestions for further improvement before the final acceptance, as follows:

  • I suggest the authors to consider avoiding the use of sentences in first person (we, our, us), please,
  • I suggest the authors reorganizing the whole Section 1. It is too long. Please, extract all background and the related work discussion from the introduction and build a separate section, a new Section 2, exclusively for the “Background and Related Work”.
  • The current Section 3 should be renamed to “Experiments and Results”.
  • The authors should proof-read the paper to enhance the overall quality of the English.
  • There is a need to include directions for future work at the end of the conclusion.

Author Response

Hello, expert

       I have made amendments according to your suggestions, as follows:

       1. All sentences with first person have been modified.
       2. Revised the chapter of the article.
       3. Correct the translation of the whole thesis.
       4. The future research direction is added.

        See the attachment for the revised article. Hope you can give us your valuable opinions. Thank you.

Reviewer 2 Report

Find attached the report in pdf format.

Author Response

Hello, expert
I have made the following modifications to your suggestions:
1. Revised the title of the article
2. Shorten the abstract
3. Modify the language of the article
4. Some shortcomings of the literature are revised.
See the attachment for the revised article. Hope you can give us your valuable opinions. Thank you

Round 2

Reviewer 2 Report

The authors have susstatially improved the manuscript and also have addressed all my concerns. I thereby, advise the publication of the manuscript. I would also like to point out a few minor typos and modifications. They are listed below.

line 61: The author has -> The authors have

line 85: one of based -> one is based

line 93: systems to be calibrated is not easy to move -> systems to be calibrated are not easy to move

line 124: central catadioptric systems is ->  central catadioptric system is 

line 152: model, in order -> model. In order

line 221:  and image distortion removed. ->  and the image distortion removed.

line 372: based on polynomial -> based on polynomial functions

line 385: introduced "Division model", -> introduced the "Division model",

line 400: non-polynomial -> non-polynomial functions

line 464: 180°; Another ->   180°; another

line 639: predicts the development direction -> predicts the possible development directions

line 663: This paper Designed -> The authors of this paper designed 

line 699:  Vertical field of view -> For the vertical field of view

line 715: a, b,ξ is the relationship between the three is simulated by MATLAB. -> The numerical relation among the three parameters a, b,ξ is simulated with MATLAB.

line 756:  this paper designs -> the authors of this paper have designed

line 785: ??, -> ??.

line 786: ,The projection -> . The projection

line 800: Back projection is from -> The back projection is performed from

line 801: , this paper discusses -> . In this paper we discuss

line 803: equation. Finally, -> equation, finally,

line 806, eq 9: ?????_???? -> ?????a_????

lines 841 and 842 should be joined together

line 847:  deviation, And ->  deviation, and

line 858: systems -> system

line 971: plane is red circle -> plane is represented by a red circle

line 1075: studied, HOVS ->  studied. HOVS

line 1078 (If I understand the phrase correctly) algorithm based on VCAM -> algorithm are based on VCAM 

line 1090 (if I understand correctly): The future work mainly focuses -> Our future work will mainly focus

line 1091: the author also focuses -> the authors will also focus

Best regards.

Author Response

Hello, expert, I have revised all the language problems you mentioned above. Please refer to the latest manuscript for details. I especially thank you for your modification of my article. Thank you again!
